# Observed decreases in on-road $CO_2$ concentrations in Beijing during COVID-19 restrictions

Di Liu[1], Wanqi Sun[2], Ning Zeng[3,4], Pengfei Han[1*], Bo Yao[2,*], Zhiqiang Liu[1], Pucai Wang[5], Ke Zheng[1], Han Mei[1], Qixiang Cai[1]

[1]Laboratory of Numerical Modeling for Atmospheric Sciences & Geophysical Fluid Dynamics, Institute of Atmospheric Physics, Chinese Academy of Sciences

[2]Meteorological Observation Centre, China Meteorological Administration, Beijing, China

[3]Department of Atmospheric and Oceanic Science, University of Maryland, USA

[4]Earth System Science Interdisciplinary Center, University of Maryland, USA

[5]Laboratory for Middle Atmosphere and Global Environment Observation, Institute of Atmospheric Physics, Chinese Academy of Sciences

*Correspondence to*: Pengfei Han (pfhan@mail.iap.ac.cn); Bo Yao (yaob@cma.gov.cn)

**Abstract:**

To prevent the spread of the COVID-19 epidemic, restrictions such as "lockdowns" were conducted globally, which led to a significant reduction in fossil fuel emissions, especially in urban areas. However, $CO_2$ concentrations in urban areas are affected by many factors, such as weather, biological sinks and background $CO_2$ fluctuations. Thus, it is difficult to directly observe the $CO_2$ reductions from sparse ground observations. Here, we focus on urban ground transportation emissions, which were dramatically affected by the restrictions, to determine the reduction signals. We conducted six series of on-road $CO_2$ observations in Beijing using mobile platforms before (BC), during (DC) and after (AC) the implementation of COVID-19 restrictions. To reduce the impacts of weather conditions and background fluctuations, we analyze vehicle trips with the most similar weather condition as possible and calculated the enhancement metric, which is the difference between the on-road $CO_2$ concentration and the "urban background" $CO_2$ concentration measured at the tower of the Institute of Atmospheric Physics (IAP), Chinese Academy of Sciences. The results showed that the DC $CO_2$ enhancement was decreased by 41 ($\pm$1.3) parts per million (ppm) and 26 ($\pm$6.2) ppm compared to those for the BC and AC trips, respectively. Detailed analysis showed that, during COVID-19 restrictions, there was no difference between weekdays and weekends during working hours (9:00-17:00 local standard time, LST). The enhancements during rush hours (7:00-9:00 and 17:00-20:00 LST) were almost twice those during working hours, indicating that emissions during rush hours were much higher. For DC and BC, the enhancement reductions during rush hours were much larger than those during working hours. Our findings showed a clear $CO_2$ concentration decrease during COVID-19 restrictions, which is consistent with the $CO_2$ emissions reductions due to the pandemic. The enhancement method used in this study is an effective method to reduce the impacts of weather and background fluctuations. Low-cost sensors, which are inexpensive and convenient, could play an important role in further on-road and other urban observations.

**Introduction:**

Since December 2019, the world has been fiercely struggling against a pandemic of a novel coronavirus named COVID-19, which was first identified in Wuhan, China (Gross et al., 2020); and then quickly identified in other countries of East Asia and Europe and the United States according to World Health Organization Novel Coronavirus (2019-nCoV) situation reports (https://www.who.int/emergencies/diseases/novel-coronavirus-2019/situation-reports). In Beijing, the first case was confirmed on 20th January 2020, followed by a quick increase in confirmed cases (SFigure 1A). From 24th January to 30th April, Beijing enacted a Level-1 response to major public health emergencies (red region in SFigure 1), and lowered the response to Level-2 from 30th April to 6th June, after "zero growth" persisted for almost one month (yellow region in SFigure 1).

As the world faced this highly infectious pandemic without efficient medication, governments enacted similar restrictions to prevent the spread of the virus: isolating cases, enacting stay-at-home orders, forbidding mass gatherings, and closing factories and schools. These restrictions highly altered the industrial production, energy consumption and transportation volume and led to sharp emission reductions (Liu et al., 2020; Le Quere et al., 2020). As previous inventory studies estimated, by early April 2020, the global daily $CO_2$ emissions had decreased by 17% (11 to 25% for ±1σ) compared with those in 2019, and the total reduction was approximately 1048 (543 to 1638) $MtCO_2$ at the end of April (Le Quere et al., 2020). Emissions from ground transportation obviously decreased by 36% (Le Quere et al., 2020). According to Liu *et al.(2020)*, emissions in China decreased 7% from January to April 2020, with ground transportation emissions dropping abruptly by 53% in February and continuing to decrease by 26% in March (SFigure 1B and 1C). In Beijing, during the first quarter of 2020, passenger traffic volumes decreased 56%, and ground transport volumes decreased 35% according to the distance-weighted passenger and freight turnover (Han et al., 2020).

Urban areas are the main $CO_2$ sources and account for more than 70% of fossil fuel emissions (Rosenzweig et al., 2010), and $CO_2$ concentrations in urban areas are dominated by weather changes (Woodwell et al., 1973;Grimmond et al., 2002); for example, high wind speed accelerates the mixing and diffusion of $CO_2$. In addition, the carbon emission reductions (258 MtC, from Le Quere *et al.(2020)*) due to COVID-19 restrictions were relatively small compared to the $CO_2$ content in the atmosphere (860 GtC, from Friedlingstein *et al.* (2019)) and carbon uptake by vegetation (the average seasonal amplitude of the net land–atmosphere carbon flux is 41.6 GtC/yr, from Zeng *et al.(2014)*. Therefore, it is difficult to detect $CO_2$ concentration decreases in the urban areas directly from sparse ground observations (Kutsch et al., 2020; Ott et al., 2020). For example, according to the daily $CO_2$ concentrations in 2019 and 2020 recorded by the tower at the Institute of Atmospheric Physics (IAP), Chinese Academy of Sciences , even though Beijing was within the strictest control/confinement period from 10th to 14th February 2020, stable weather (in which the planetary boundary layer heights (PBLHs) were only ~600 m) led to $CO_2$ concentrations that were approximately 90 parts per million (ppm) higher than those on the same date in 2019 (PBLHs were ~ 900 m)(SFigure 1D). Sussmann and Rettinger (2020) also proved it. Despite global emission reductions due to COVID-19 restrictions, they found a historic record high in column-averaged atmospheric carbon dioxide (XCO2) in April 2020 by using Total Carbon Column Observation Network (TCCON) data. By assuming that the COVID-19-related $CO_2$ growth rate reduction of 0.32 $ppm/yr^2$ in 2020 at Mauna Loa is true and measured (from the UK Met Office; an overall 8% emission reductions in 2020), they found that there is a ~0.6 year 'delay' to separates TCCON-measured growth rates and the reference forecast (absence of COVID-19 restrictions).

With the knowledge that urban ground transportation was strongly suppressed due to COVID-19 restrictions, we designed on-road observations by using a mobile platform to detect reduction signals. These observations could provide $CO_2$ data with higher spatiotemporal resolution than satellite and ground observations and have been widely used for carbon monitoring in urban and suburban areas (for instance, on-road $CO_2$ concentration distributions were presented as transects in urban areas along routes) (Idso et al., 2001;Bush et al., 2015;Sun et al., 2019). Almost all studies agreed that weather (for example, wind

speed, which is directly associated with $CO_2$ mixing and dilution) is a dominant factor and should be considered during analysis. Reducing the impact of weather is still a problematic. On the other hand, examining the enhancement, which is the calculated the difference in the $CO_2$ concentration between urban and rural background observations, could effectively

reduce the influence of background $CO_2$ fluctuations, and this metric has been widely used for monitoring urban carbon emissions and $CO_2$ concentrations (Idso et al., 1998;Idso et al., 2002;George et al., 2007;Mitchell et al., 2018;Perez et al., 2009).

To determine the $CO_2$ concentration reduction "signal" due to decreased ground transportation emissions during COVID-19

restrictions, we chose the most similar weather conditions as possible and calculated the enhancements metric by subtracting the "baseline" IAP tower $CO_2$ concentration from the observed on-road $CO_2$ concentration to reduce impact of background $CO_2$ fluctuations. Our results may provide direct evidence of ground transportation emission reductions due to COVID-19 restrictions, and this method could be an appropriate tool to analyse the $CO_2$ concentration and emissions related to urban ground transportation in future works.

**Methods and Data:**

We conducted six on-road observations in Beijing using mobile platforms before (BC; 1 trip: 20[th] February 2019), during (DC; 4 trips: 13[th], 20[th], 21[st] and 22[nd] February 2020) and after (AC; 1 trip: 9[th] May 2020) COVID-19 restrictions (vertical lines in SFigure 1 indicate the trip dates). These trips covered four ring roads that circled the city: the 2[nd] (with length of 33

100    km), 3[rd] (48 km), 4[th] (64 km) and 5[th] (99 km) ring roads, from innermost to outermost, as shown in Figure 1. All trips were conducted during the daytime; four of them were on weekdays and two others were on a Saturday. Four trips covered at least one rush hour (7:00-9:00 local standard time (LST) for morning rush hour; 17:00-20:00 LST for evening).

To reduce the influence of background $CO_2$ fluctuations, we chose the similar weather conditions. As shown in Table 1, four

aspects were considered: (1) real-time panoramic photographs collected from the IAP tower (photograph available from: http://view.iap.ac.cn:8080/imageview/); (2) the $PM_{2.5}$ (atmospheric particulate matter with a diameter of less than 2.5 μm) concentration from the Olympic Sports Center Station (40.003 °N, 116.407 °E, 5 m height, purple square in Figure 1A), which is run by the Ministry of Ecology and Environment of China (Zhang et al., 2015); (3) wind speed data (collected from: https://www.wunderground.com/history/daily/cn/beijing/ZBNY/date/2020-5-9) ; and (4) PBLH data, which are related to

vertical mixing and diffusion of pollution/$CO_2$ emitted near the ground (Su et al., 2018). These data were collected from National Centers for Environmental Prediction Global Forecast System (GFS) reanalysis dataset (resolution: $0.25° \times 0.25°$), which is a globally gridded dataset representing the state of the Earth's atmosphere and incorporating observations and numerical weather prediction model output.

Then, on-road $CO_2$ concentration enhancements were calculated by subtracting the simultaneous $CO_2$ concentrations detected at the IAP tower, which served as the "baseline" for Beijing city (Eq. 1).

$$CO_2 \text{ enhancement} = CO_2 \text{ (on-road)} - CO_2 \text{ (IAP tower)} \qquad \text{(Eq. 1)}$$

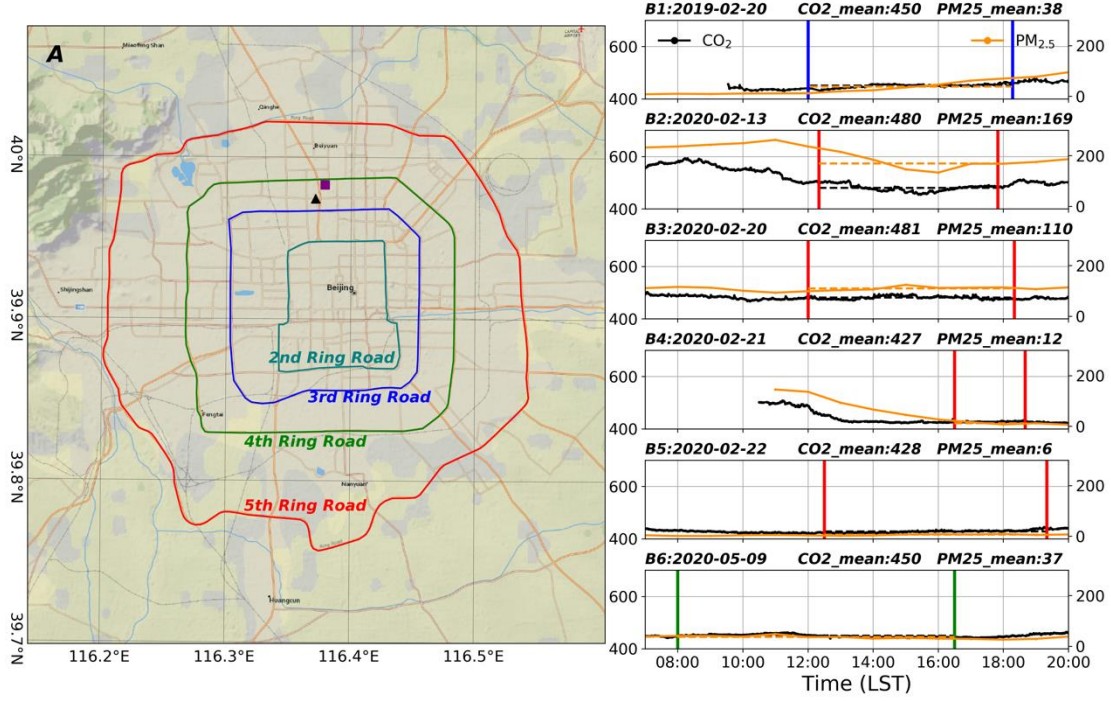

Figure 1. A: The locations of the $2^{nd}$, $3^{rd}$, $4^{th}$ and $5^{th}$ ring roads, the IAP tower (black triangle) and Olympic Sports Centre station (purple square); B1-B6: $CO_2$ concentration at the IAP tower and $PM_{2.5}$ concentration data from the Olympic Sports Center station during six trips.

Table 1. Weather conditions during six trips.

| Label/date | Weather condition | Air condition (PM2.5: μg/m³) | Wind speed (m/s) | PBLH (m) | Real-time panoramic photographs |
|---|---|---|---|---|---|
| BC 2019-2-20 (Wed) | Clear day | 38 | 2.5 | 897.7 | |
| DC 2020-2-13 (Fri) | Heavily polluted day | 169 | 2.5 | 589 | |
| DC 2020-2-20 (Fri) | Lightly polluted day | 110 | 1.3 | 691 | |

| DC 2020-2-21 (Fri) | Clear day | 12 | 2.5 | 1587 | |
|---|---|---|---|---|---|
| DC 2020-2-22 (Fri) | Clear day | 6 | 3.6 | 1113 | |
| AC 2020-5-9 (Sat) | Clear day | 37 | 1.6 | 608 | |

**CO$_2$ concentration at IAP tower:**

The IAP tower is a 325 m-high meteorological tower located at 116.3667 °E, 39.9667 °N, 49 m above sea level in northwest Beijing (Figure 1, black triangle) (Cheng et al., 2018). The CO$_2$ concentration was determined at three levels in this study: surface level (~2 m above the ground), lower level (~80 m) and upper level (~280 m). The CO$_2$ concentrations were

130 measured by a Picarro G2301 greenhouse gas concentration analyser (Picarro, 2019).The instrument was calibrated by using standard gas for every 3 hours, and each calibration lasted 5 minutes. The standard gasses were from the Meteorological Observation Center of the China Meteorological Administration (MOC/CMA) and were traced to the World Meteorological Organization (WMO) X2007 scale. The measurement accuracy was ~0.1 ppm. The CO$_2$ concentration was recorded by every 2 seconds, and these data were averaged into 1-minute intervals. Before 2020 (including the trip on 20[th] February

2019), the CO$_2$ concentration was measured at the lower and upper levels alternately for every 5 minutes, and the measurement at each level lasted 5 minutes. After 2020 (including the other 5 trips), the CO$_2$ concentration was continuously measured at the surface level. To maintain consistency as much as possible, we used the lower-level CO$_2$ before 2020 and the surface level CO$_2$ after 2020.

**On-road CO$_2$ concentration data:**

Three different CO$_2$-observing instruments were carried by vehicles during six on-road trips (Table 2).

1) On 20[th] February 2019, a Picarro G2401 (Picarro, 2017) was installed on a vehicle; the air intake was set on the roof of the vehicle to avoid contact with direct plumes emitted from surrounding cars. The intake was linked/connected through a 2 m pipe with a particulate matter filter to the Picarro system (Figure 2A and 2B). The instrument characteristics and

145 accuracy have been described by Sun *et al.* (2019). The CO$_2$ concentrations were collected every 2 seconds and then averaged into 1-minute intervals.

2) During COVID-19 restrictions (surveys on 13[th], 20[th], 21[st] and 22[nd] February 2020), a LI-COR LI-7810 CH$_4$/CO$_2$/H$_2$O trace gas analyser was adopted, which uses optical feedback-cavity enhanced absorption spectroscopy (LI-COR, 2019). This instrument could obtain a CO$_2$ concentration with a precision of 3.5 ppm for 1 second and within 1 ppm after

150 1-minute averaging (laboratory testing). The observation platform of the LI-7810 was similar to that of the Picarro system. Before departure, the instrument was calibrated by using standard calibration gas to correct the drift.

3) On 9[th] May 2020, a low-cost light sensor was adopted and installed on the front windshield of the vehicle (Figure 2C). The instrument mainly consisted of three non-dispersive infrared (NDIR) $CO_2$ measurement sensors (named K30), and one environment (temperature, humidity and pressure) sensor (named BME). Although the original precision of each K30 was ±30 ppm, after calibration and environmental correction in the laboratory and before departure, the accuracy was improved to within ±5 ppm comparing with Picarro (Martin et al., 2017;SenseAir, 2019). Here, we used three K30s in one instrument to recognize and eliminate data anomalies and used the average $CO_2$ concentrations from the three K30s for analysis. Figure 3 shows the details of the experiment conducted on 22[nd] February 2020, for which one low-cost light sensor and Picarro were installed on the same vehicle for on-road monitoring. The results showed that the low-cost light sensor results were highly consistent with those of the Picarro system, with root mean square errors (RMSEs) less than 5 ppm.

*Table 2. Instrument parameters for six on-road observations*

| Label | Date | Instrument | Accuracy | Temporal resolution (original->processed) |
|-------|------|-----------|----------|------------------------------------------|
| BC | 2019-2-20 | Picarro G2401 | ±0.1 ppm | 2 seconds -> 1 minute |
| DC | 2020-2-13 | LI-COR LI-7810 | ±3.5 ppm (for 1 second); improved into ±1 ppm (for 1 minute) | 1 second -> 1 minute |
|  | 2020-2-20 | LI-COR LI-7810 |  |  |
|  | 2020-2-21 | LI-COR LI-7810 |  |  |
|  | 2020-2-22 | LI-COR LI-7810 |  |  |
| AC | 2020-5-9 | Low-cost Sensor (K30) | ±5 ppm | 2 seconds -> 1 minute |

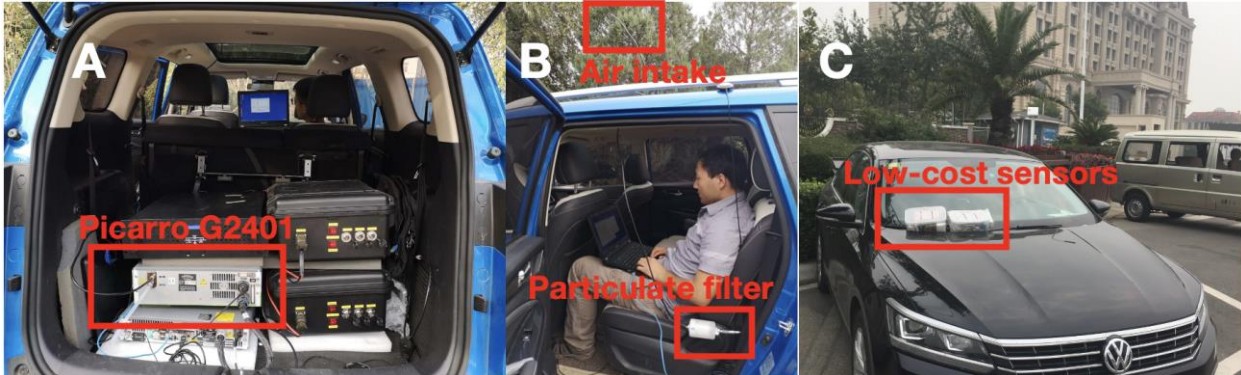

*Figure 2. Photographs of the instrument installation for the on-road observations. (A) and (B) Picarro system installed in the vehicle; (C) low-cost non-dispersive infrared (NDIR) sensors installed on the front windshield of the vehicle.*

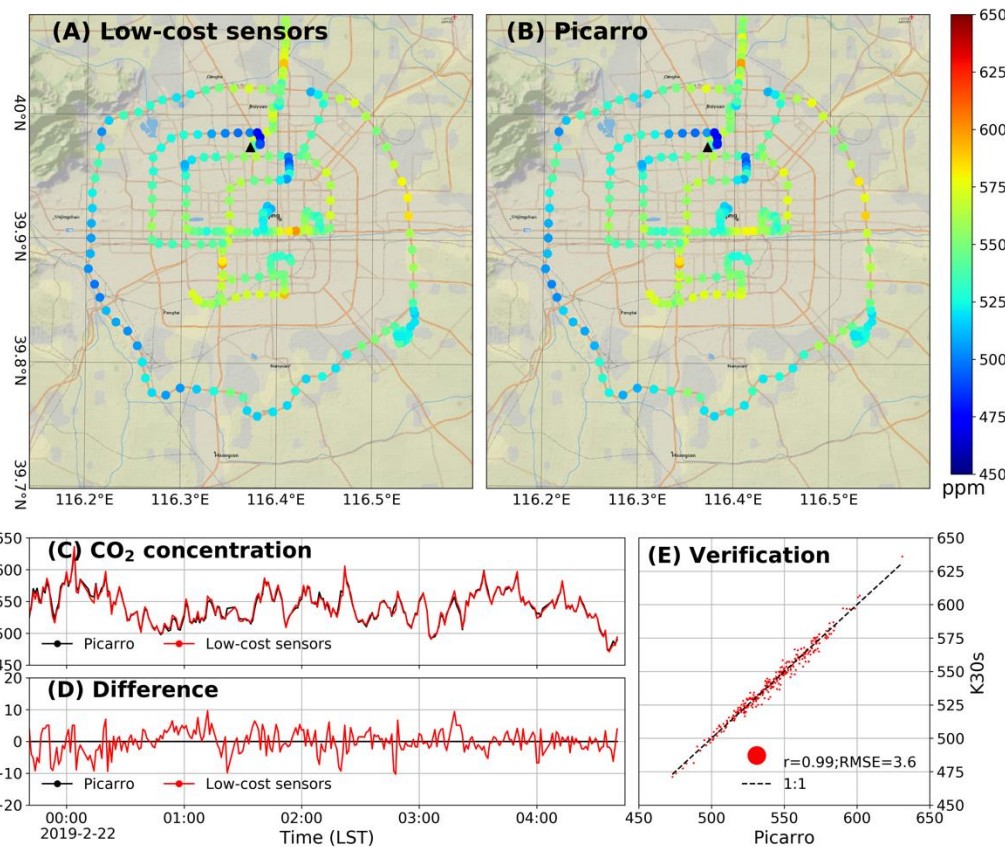

Figure 3. Verification of low-cost sensors for on-road observations. (A): Map of $CO_2$ concentrations measured by the low-cost sensor; (B): map of the $CO_2$ concentration measured by the Picarro system on the same vehicle; (C): time series of the $CO_2$ concentrations measured by the low-cost sensor and Picarro system; (D): difference (low-cost sensor concentration minus Picarro concentration); (E): scatter plot of the low-cost sensor and Picarro data, with an RMSE of 3.6 ppm.

**Auxiliary data and analysis:**

The global positioning system (GPS) data for BC and DC were collected by a GPS receiver (BS-70DU) (Sun et al., 2019). For AC, the data were collected by using mobile software (GPS Tracks), which provided time, longitude, latitude, speed and altitude at 1-second resolution. These geographic information data were averaged into 1-minute intervals and then matched with the $CO_2$ concentration data according to time.

Two remote sensing images were adopted (captured on 21[st] February 2019 at 11:40:00 LST from a Google Earth image, with 0.37 m spatial resolution; 19[th] February 2020 at 10:20:08 LST from a Beijing-2 remote sensing satellite panchromatic image, with 0.8 m spatial resolution). Considering the availability of data, we used the images from the closest date and only part of the urban area. The comparison region covered 10 km of the 3[rd] Ring Road (accounting for 21 % of the whole road) and 13.4 km of the 4[th] Ring Road (also 21 % of the whole road). We used a visual interpretation method to obtain the numbers of vehicles on the 3[rd] and 4[th] Ring Roads for BC and DC, respectively.

To understand the traffic situation, we also collected the real-time traffic congestion conditions (for each road), road name, geographic information, road type and average speed as one-hour data from the Autonavi Open Platform (https://lbs.amap.com/).

**Results:**

**On-road CO$_2$ concentration:**

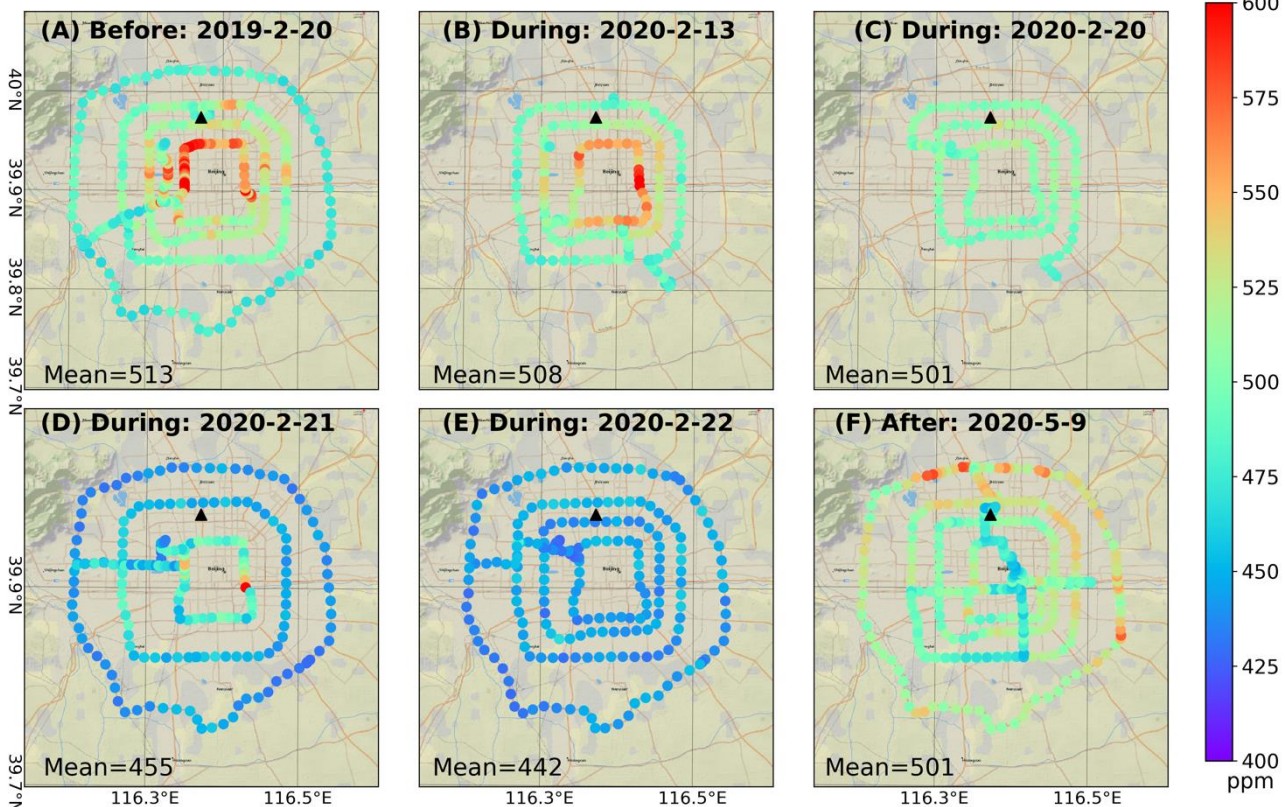

Figure 4. CO$_2$ concentration maps for six on-road trips. Circles are the locations of CO$_2$ concentration records taken at a 1-minute interval (see methods). All subplots have the same colour scale, ranging from 400 to 600 ppm. The black triangle is the location of the IAP tower. One trip (A:20$^{th}$ February 2019) was conducted before the COVID-19 restrictions, with an average of 513 (with an instrument uncertainty of ±0.1) ppm. Four trips (B-E: 13$^{th}$, 20$^{th}$, 21$^{st}$ and 22$^{nd}$ February 2020) were conducted during COVID-19 restrictions, with averages of 508 (±1), 501 (±1), 455 (±1) and 442 (±1) ppm, respectively. One trip (F: 9$^{th}$ May 2020) was conducted after the COVID-19 restrictions, with an average of 501(±5) ppm.

The CO$_2$ concentration maps of six on-road trips are shown in Figure 4. According to Table 1, we selected four trips as the trips with the most similar weather conditions: one BC trip (20$^{th}$ February 2019, Figure 4A), two DC trips (21$^{st}$ and 22$^{nd}$ February 2020, Figure 4D and 4E) and 1 AC trip (9$^{th}$ May 2020, Figure 4F). Statistically, the average of the 2 DC trips was 444 (±1) ppm, which was 69 (±1.1) and 57 (±6) ppm lower than that of the BC and AC trips, respectively. The other two DC trips (13$^{th}$ and 20$^{th}$ February) were conducted on (lightly/heavily) polluted days, and the CO$_2$ concentrations on these two days were as high as those during the BC and AC trips.

We chose one DC trip (21$^{st}$ February 2020) for further analysis and compared it to the BC and AC trips. All three trips were conducted on clear days, and their trajectories were similar, from the outermost circle to the innermost circle, and covered one (morning or evening) rush hour. The difference was that the BC and DC trips hit the evening rush hour on the innermost ring road, whereas the AC trip hit the morning rush hour on the outermost ring road. This difference explained why the CO$_2$ concentration was high on the innermost road (2$^{nd}$ Ring Road) in figure 4A and 4D and on the outermost road (5$^{th}$ Ring Road) in figure 4F. The comparison of the three trips indicated that the CO$_2$ concentration in Figure 4D was lower than those in Figure 4A and 4F, and the statistics show that the mean CO$_2$ of the DC trips was approximately 58 (±1.1) and 46 (±6) ppm lower than those of the BC and AC trips, respectively. In addition, the average CO$_2$ concentration observed at the IAP tower

during the same periods was much lower than the on-road observations (Figure 1B). These concentration differences (gradients) also implied that ground transportation emissions were a major $CO_2$ source on these urban roads.

However, it was difficult to completely eliminate the impact of background $CO_2$ fluctuations only though selecting trips with the most similar weather conditions. For example, the PBLHs during two DC trips with the most similar weather were 1587 m and 1113 m, which were almost twice of those during the BC and AC trips (Table 1). The $CO_2$ concentrations at the IAP tower also indicated that during these two DC trips, the $CO_2$ concentrations were 427 ($\pm 0.1$) and 428 ($\pm 0.1$) ppm, which were approximately 20 ppm lower than those for the BC and AC trips (in Figure 1).

**On-road $CO_2$ enhancement:**

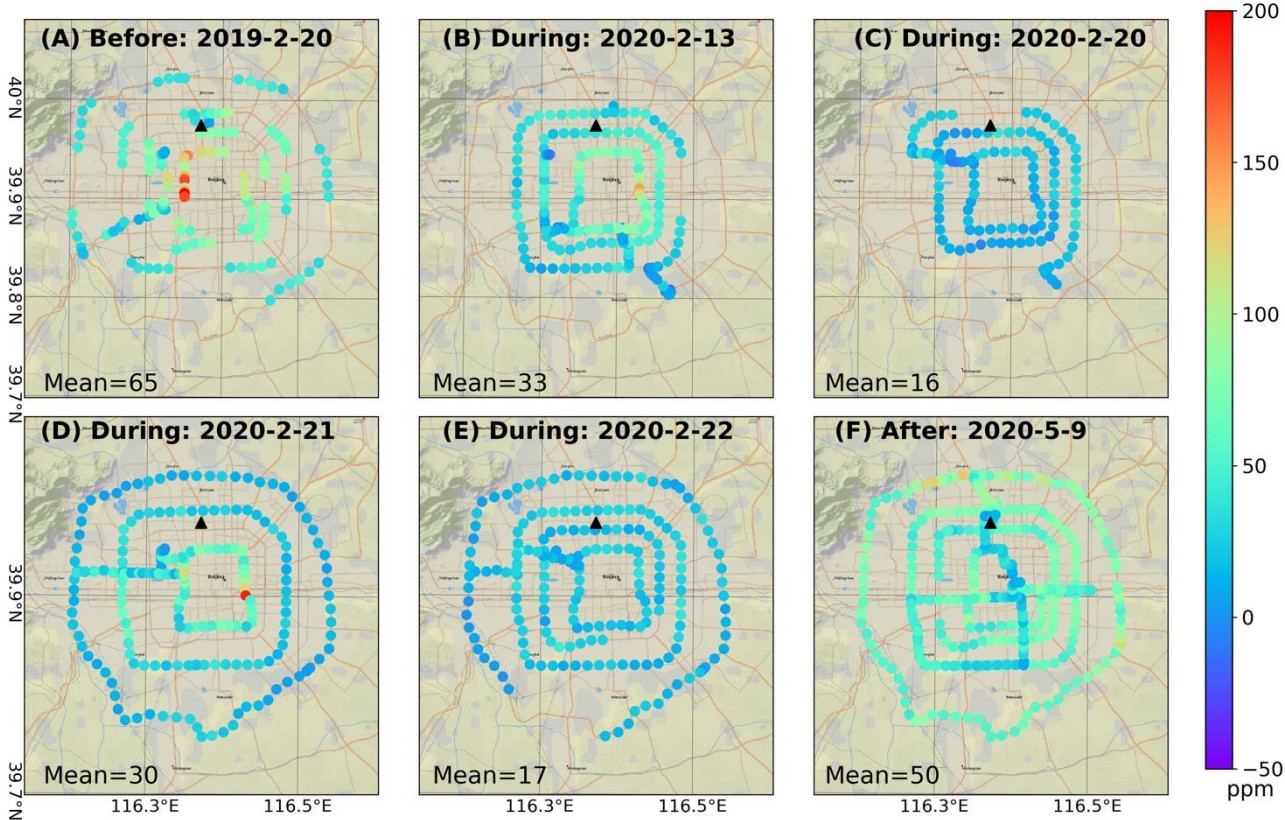

*Figure 5. Maps of the $CO_2$ enhancement for all six trips calculated by subtracting the IAP tower measurements from the on-road $CO_2$ measurements matched temporally. All subplots have the same colour scale, ranging from -50 to 200 ppm. One trip (A: 20th February 2019) was conducted before the COVID-19 restrictions, with an average of 65 ($\pm 0.2$) ppm. Four trips (B-E: 13th, 20th, 21st and 22nd February 2020) were conducted during the COVID-19 restrictions, with averages of 33 ($\pm 1.1$), 16 ($\pm 1.1$), 30 ($\pm 1.1$) and 17 ($\pm 1.1$) ppm, respectively. One trip (F: 9th May 2020) was conducted after the COVID-19 restrictions, with an average of 50 ($\pm 5.1$) ppm.*

To further reduce the influence of background $CO_2$ variations, we calculated the $CO_2$ enhancement for six trips by subtracting the $CO_2$ concentration at IAP tower from the on-road $CO_2$ concentration (shown in Figure 5). The spatial distribution patterns of the enhancement were similar to the distribution of the CO2 concentration maps, in which the enhancements during rush hours were much higher for all trips. Furthermore, the refined spatial distribution of the $CO_2$ gradient implied emissions from ground transportation.

It is worth noting that the enhancements for the four DC trips were almost the same, although the weather conditions (based on the PBHL, PM$_{2.5}$ and wind speed data) during these trips were quite different. However, the DC enhancements were obviously different from the BC and AC enhancements. During the two DC trips on polluted days (13$^{th}$ and 20$^{th}$ February 2020), the mean CO$_2$ concentrations were similar to those during the BC and AC trips (Figure 4B and 4C); however, the enhancements extracted the traffic emission signals from the background, with averages of 33 ($\pm$1.1) and 16 ($\pm$1.1) ppm (Figure 5B and 5C). Statistically, the average of the four DC enhancements was 24 ($\pm$1.1) ppm, which was 41 ($\pm$0.2) and 26 ($\pm$6.2) ppm lower than those of the BC and AC enhancements.

**Diurnal variation analysis:**

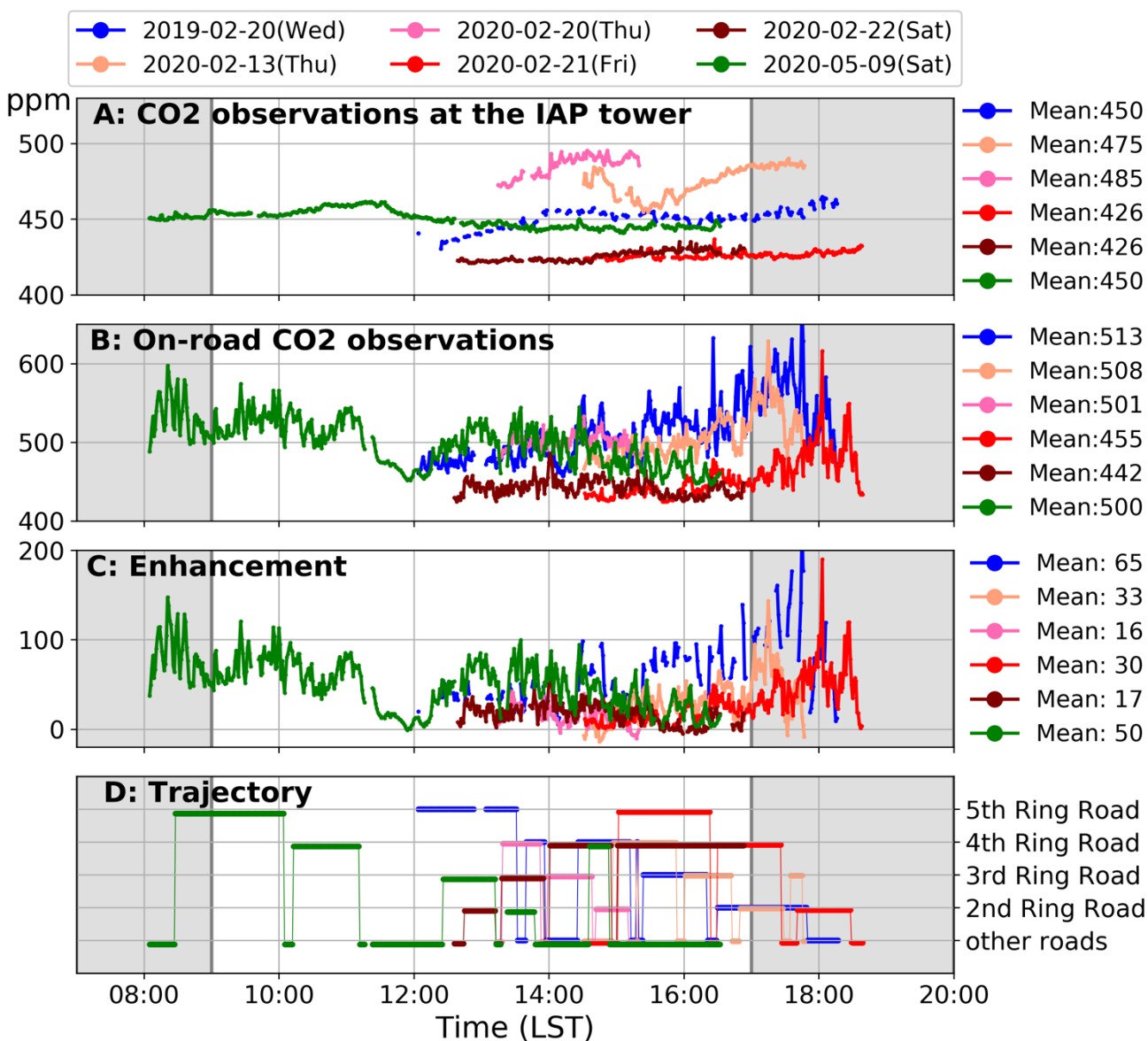

*Figure 6. The six trips were plotted on a single day. The two grey regions refer to the morning and evening rush hours. The six colourful lines represent the six trips on different days. Four of the 6 trips covered at least one (morning/evening) rush hour. Panel A shows the CO$_2$ concentration at the IAP tower during the trips. Panel B shows the on-road CO$_2$ concentration. Panel C shows the CO$_2$ enhancements. Panel D shows the six trip trajectories.*

Figure 6 shows the diurnal variation in the CO$_2$ concentrations from IAP tower observations, on-road CO$_2$ concentrations, enhancements and trajectories. In Figure 6A, the CO$_2$ concentration measured at the IAP tower were stable and showed an approximate 50 ppm difference between trips. The CO$_2$ concentrations at the IAP tower during the first two DC trips (13$^{th}$

and 20[th] February 2020) were ~30 ppm higher than those during the BC and AC trips. However, the $CO_2$ concentrations during the other two DC trips (21[st] and 22[nd] February 2020) were ~20 ppm lower than those during the BC and AC trips. These "baseline" $CO_2$ concentration fluctuations make the on-road observations not directly comparable. In Figure 7B, the $CO_2$ concentrations show a "double-peak" pattern, with peaks during the morning (7:00-9:00) and evening (17:00-20:00) rush hours. During the rush hours, the $CO_2$ concentrations ranged from 500 to 600 ppm, which were approximately 100 ppm higher than the concentrations during working hours (9:00-17:00). The comparison of BC and AC indicates that the $CO_2$ concentrations measured on 13[th] and 20[th] February 2020 did not significantly decrease during 12:00-17:00. However, the $CO_2$ concentrations measured on 21[st] and 22[nd] February 2020 were much lower (~50 ppm) than those measured during the BC and AC trips. This difference is consistent with the spatial distribution mentioned before and is most likely due to background $CO_2$ fluctuations.

In Panel C, all DC enhancements were generally lower than the BC and AC enhancements, and the statistics for different time periods are listed in Table 3. However, we also found small enhancements for BC and AC, similar to thoese for DC. For example, the AC enhancement at 12:00-16:00 was almost the same as the DC enhancement at that time. By examining the trip routes (Panel D), we found that during that period, the on-road observation vehicle was not driving on the main ring roads. As another example, the BC enhancement at 18:00 indicates that the enhancement decreased in a stepwise manner, also because the vehicle drove on other roads (Panel D).

Table 3. $CO_2$ enhancement (mean and instrumental uncertainties) for six trips over different periods (ppm)

| Label | Observation date | Weather condition | Total average (07:00-20:00) | Morning rush hours (07:00-09:00) | Working hours (09:00-17:00) | Evening rush hours (17:00-20:00) |
|---|---|---|---|---|---|---|
| BC | 2019-2-20 (Wed) | Clear | 65 (±0.2) | - | 54 (±0.2) | 100 (±0.2) |
| DC | 2020-2-13 (Thu) | Stable/heavy pollution | 33 (±1.1) | - | 26 (±1.1) | 55 (±1.1) |
| | 2020-2-20 (Thu) | Stable//light pollution | 16 (±1.1) | - | 16 (±1.1) | - |
| | 2020-2-21 (Fri) | Windy day | 30 (±1.1) | - | 16 (±1.1) | 50 (±1.1) |
| | 2020-2-22 (Sat) | Windy day | 17 (±1.1) | - | 17 (±1.1) | - |
| AC | 2020-5-9 (Sat) | Windy day | 50 (±5.1) | 80 (±5.1) | 46 (±5.1) | - |
| | Total BC-DC | | 41 (±1.3) | - | 35 (±1.3) | 48 (±1.3) |
| | Total AC-DC | | 26 (±6.2) | - | 27 (±6.2) | - |

The mean enhancement for the whole BC trip was 65 (±0.2) ppm, and the average for the evening rush hours (100 ±0.2 ppm) was two times that for the working hours (54 ±0.2 ppm). This result implies that the increase in vehicle volume during the evening rush hours leads to large traffic emissions and an increase in the on-road $CO_2$ concentration. For DC, all trips

covered the working hours, with a low enhancement of approximately 20 ppm. There was no obvious difference between weekdays and weekends during working hours. The reason may be that the government encouraged people to work remotely at home. Therefore, even on weekdays, according to traffic conditions, the commute volume was low (SFigure 2). Among these four trips, two (on 13[th] and 20[th] February 2020) covered the evening rush hours with high averaged enhancements of 55 ($\pm$1.1) and 50 ($\pm$1.1) ppm. Therefore, the total average enhancements for these two trips were higher than those for the other two trips, which covered only working hours. For AC, on 9[th] May 2020, although it was a Saturday, many residents chose to go out of town for the weekends. The morning rush hours still existed, with a high enhancement of 80 ($\pm$5.1) ppm, and then during the working hours, the enhancement decreased to 46 ($\pm$5.1) ppm.

The comparison of trips showed that the average $CO_2$ enhancement for the 4 DC trips was 41 ($\pm$1.3) and 26 ($\pm$6.2) ppm lower than that for the BC and AC trips, respectively. The average AC enhancement was 15 ($\pm$5.3) ppm lower than the average BC enhancement. This difference may be caused by two factors: 1) The first relates to "weekly effects"; a previous study also suggested that, compared to that during weekdays, the average daily traffic $CO_2$ emissions during weekends in the north part of the fifth Ring Road (LinCui Road - Anli Road, 3 km) decreased by 5% in 2014 (Zheng et al., 2020). 2) Until 9[th] May 2020, although there were approximately 30 days without increases in COVID-19 cases in Beijing, the city was still under Level-2 response control; social life was recovering but had not yet completely recovered.

**Analysis of $CO_2$ enhancement for independent time periods and roads:**

According to the previous analysis, we found that enhancement exhibited a strong correlation with time (rush or working hours) and road type. Therefore, we statistically analysed the $CO_2$ enhancement according to the road type and time period, as shown in Figure 7. In Figure 7A, on 13[th] and 20[th] February 2020, the $CO_2$ concentrations on the other, 2[nd], and 4[th] ring roads and all roads were at the same levels as those during the BC and AC trips. However, in Figure 7B, the four DC enhancements were generally lower than those during AC and BC for all road types. Although on the 2[nd] Ring Road, the DC enhancements on 13[th] and 21[st] February 2020 were almost the same as the BC and AC enhancements, the DC trips were during rush hours, whereas the AC and BC trips were during working hours. Some very high deviations also occurred (rush hours on the other roads 2[nd] and 5[th] ring roads), which indicates the dispersion of the $CO_2$ enhancement. The reason for this difference is that we classified all roads excluding the ring roads as other roads, which may have included arterial and residential roads, so the different road types may have increased the deviation. For the 2[nd] and 5[th] ring roads, high deviation occurred because during rush hour, traffic flow and transportation varied greatly and resulted in drastic changes in the $CO_2$ enhancement, which also caused much higher deviations. After a statistical significance test, we found that the $CO_2$ enhancement difference between working times and rush hours for all trips was significant ($p < 0.02$, assuming that $\alpha$=0.05). The $CO_2$ enhancement for BC was also significantly different from that for DC ($p < 0.05$); however, the difference between the AC and BC enhancements was not significant. This suggests that the decreased $CO_2$ enhancement observed during the COVID-19 restrictions was significantly different from those before and after the COVID-19 restrictions. We also calculated specific statistics, which are listed in Table 4.

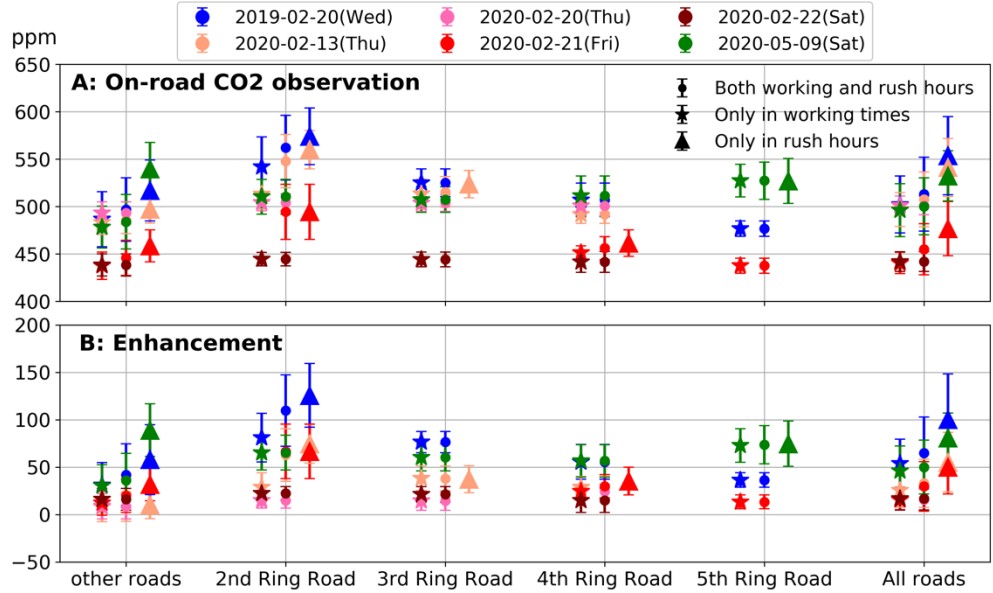

Figure 7. *Statistical analysis (mean and one standard deviation) of all on-road trips according to the road types and times. Panel A shows the on-road CO$_2$ concentration. Panel B shows the CO$_2$ enhancement.*

Table 4. Statistical analysis (mean/one standard deviation) of the CO$_2$ enhancement for six trips according to the time and road type

| Label | Date | Time | Other roads | 2nd Ring Road | 3rd Ring Road | 4th Ring Road | 5th Ring Road | All roads | Significance test (p) | |
|---|---|---|---|---|---|---|---|---|---|---|
| | | | | | | | | | Working hours compared to rush hours | DC/AC compared to BC |
| BC | 2019-2-20 (Wed) | Working hours | 31/24 | 81/26 | 77/11 | 56/18 | 37/8 | 54/26 | 0.015 | - |
| | | Rush hours | 58/37 | 125/34 | - | - | - | 100/48 | | - |
| | | Both | 42/33 | 109/38 | 77/11 | 56/18 | 37/8 | 65/38 | - | - |
| DC | 2020-2-13 (Thu) | Working hours | 8/16 | 29/15 | 38/13 | 29/11 | - | 26/18 | 0.018 | - |
| | | Rush hours | 10/14 | 74/20 | 37/14 | - | - | 55/31 | | - |
| | | Both | 9/16 | 63/28 | 38/13 | 29/11 | - | 33/26 | - | 0.041 |
| | 2020-2-20 (Thu) | Working hours | 9/13 | 15/8 | 14/10 | 24/8 | - | 16/11 | - | - |
| | | Rush hours | - | - | - | - | - | - | | - |
| | | Both | 9/13 | 15/8 | 14/10 | 24/8 | - | 16/11 | - | 0.001 |
| | 2020-2-21 (Fri) | Working hours | 12/13 | - | - | 25±7 | 13±7 | 16±10 | 0.002 | - |
| | | Rush hours | 32/17 | 67/29 | - | 35/15 | - | 50/28 | | - |
| | | Both | 20/18 | 67/29 | - | 30/13 | 13/7 | 30/26 | - | 0.026 |
| | 2020-2-22 | Working | 16/11 | 22/7 | 21/8 | 15/13 | - | 17/12 | - | - |

| | | | | | | | | | | | |
|---|---|---|---|---|---|---|---|---|---|---|---|
| | (Sat) | hours | | | | | | | | | |
| | | Rush hours | - | - | - | - | - | - | | - | |
| | | Both | 16/11 | 22/7 | 21/8 | 15/13 | - | 17±12 | - | 0.001 | |
| AC | 2020-5-9 (Sat) | Working hours | 30/22 | 65/18 | 60/14 | 57/17 | 73/18 | 46/26 | 0.008 | - | |
| | | Rush hours | 89/28 | - | - | - | 75/24 | 81/26 | | - | |
| | | Both | 36/29 | 65/18 | 60/14 | 57/17 | 73/20 | 50/28 | - | 0.41 | |

325

## Discussion:

### Analysis of the correlation with traffic flow:

It was difficult to obtain a quantitative evaluation of the influence of COVID-19 restrictions on $CO_2$ emissions from traffic because of limited data. In this study, we found that the one-trip enhancement for DC (on 21[st] February 2020, with weather conditions and a route that were the most similar to those for the BC and AC trips) was 30 ($\pm1.1$) ppm. The enhancement accounted for 46% of that for BC (65 $\pm0.2$ ppm), and the enhancement for AC (50 $\pm5.1$ ppm) accounted for 77% of that for BC. Here, we adopted four datasets and methods to explain our hypothesis that the decrease in traffic volume led to a reduction in on-road $CO_2$ emissions and concentration during the COVID-19 restrictions. First, according to the "analysis of road traffic operation in Beijing during COVID-19 in 2020" published by the Beijing Transport Institute, during the first 8 weeks (from 1[st] February to 31[st] March, the DC period in this study), the Beijing ground transportation index (calculated based on the ratio of congested road length to the whole road length) decreased by 53% compared to that on normal days, whereas, during 1[st] April to 31[st] May, the index recovered to 92% (Zhang, 2020). The index implied that traffic flow for DC was dramatically decreased compared to that for BC, and the index for AC almost recovered but not completely. This index variation is consistent with our observations. Second, two remote sensing images from similar dates were adopted (Figure 8). According to statistics and estimations based on coverage area, we found that the BC traffic flows on the main roads of the 4[th] and 3[rd] Ring Roads were 227 and 226 veh/km (vehicles per kilometre), respectively. However, the DC traffic flow decreased to 35 and 34 veh/km, reflecting a reduction of approximately 85%. With the assuming that emission factors were the same, the $CO_2$ emissions on roads for DC may have sharply decreased by approximately 85% compared to those for BC. This difference is higher than the passenger transportation decrease estimated by Han *et al.* (2020) (56% in the first quarter of 2020) because remote sensing images are snapshots and cover only part of the urban area. Moreover, Hans' results are the average of the first three months and the entire Beijing administrative region. Third, we also used traffic congestion condition data, although with low temporal and spatial resolution, to indicate the on-road traffic flow and emissions (Figure 9). Fourth, the vehicle speed maps of the six trips were plotted (Figure 10). Overall, these maps reflect the spatial patterns of road traffic conditions during the surveys and could also reflect the specifics on a single road. However, these maps are sensitive to subjective speed variations caused by drivers, such as when facing traffic lights.

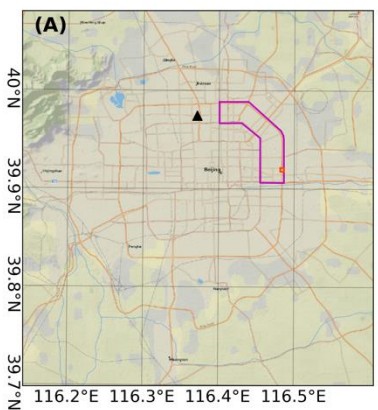
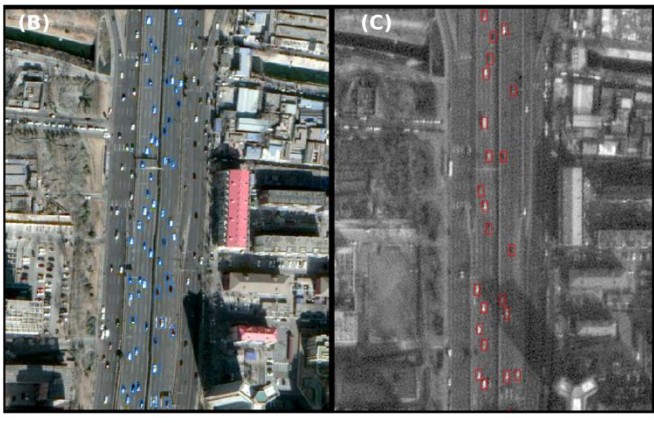

*Figure 8. Traffic volume comparison with using remote sensing images. (A) Coverage region of remote sensing images (purple polygon) and example region shown on the right (red square); (B) remote sensing images from Google Earth on 21st February 2019 at 11:42:00 (LST), with a spatial resolution of 0.37 m for multispectral band images; 61 vehicles on the main road were interpreted (labelled by blue polygons); (C) remote sensing image from the Beijing-2 satellite on 19th February 2020 at 10:20:08 (LST), with a spatial resolution of 0. 8 m for the panchromatic band images and 24 vehicles labelled with red polygons.*

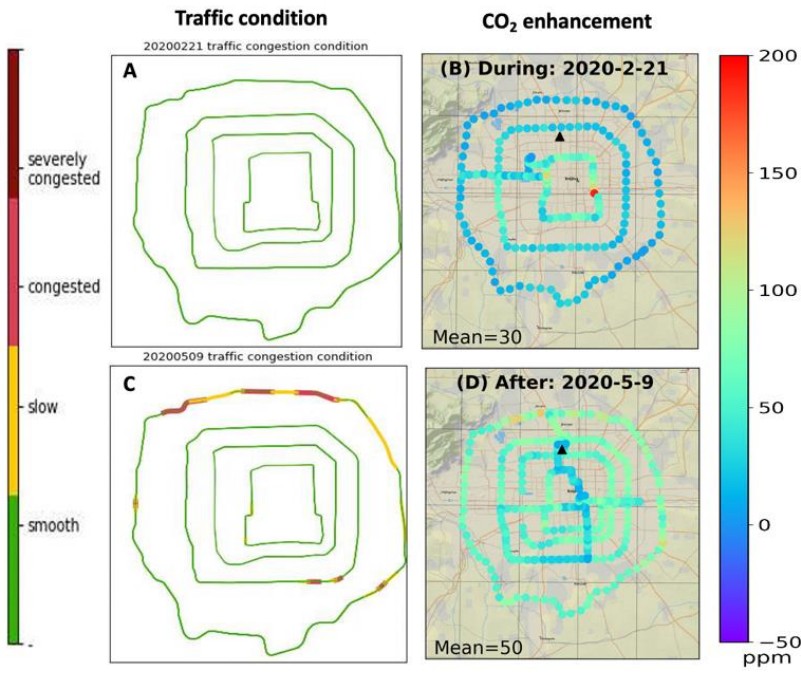

*Figure 9. Comparison of traffic conditions with the $CO_2$ enhancement. (A) Traffic conditions on 21st February 2020; (B) $CO_2$ enhancement on 21st February 2020; (C) traffic conditions on 9th May 2020; (D) $CO_2$ enhancement on 9th May 2020.*

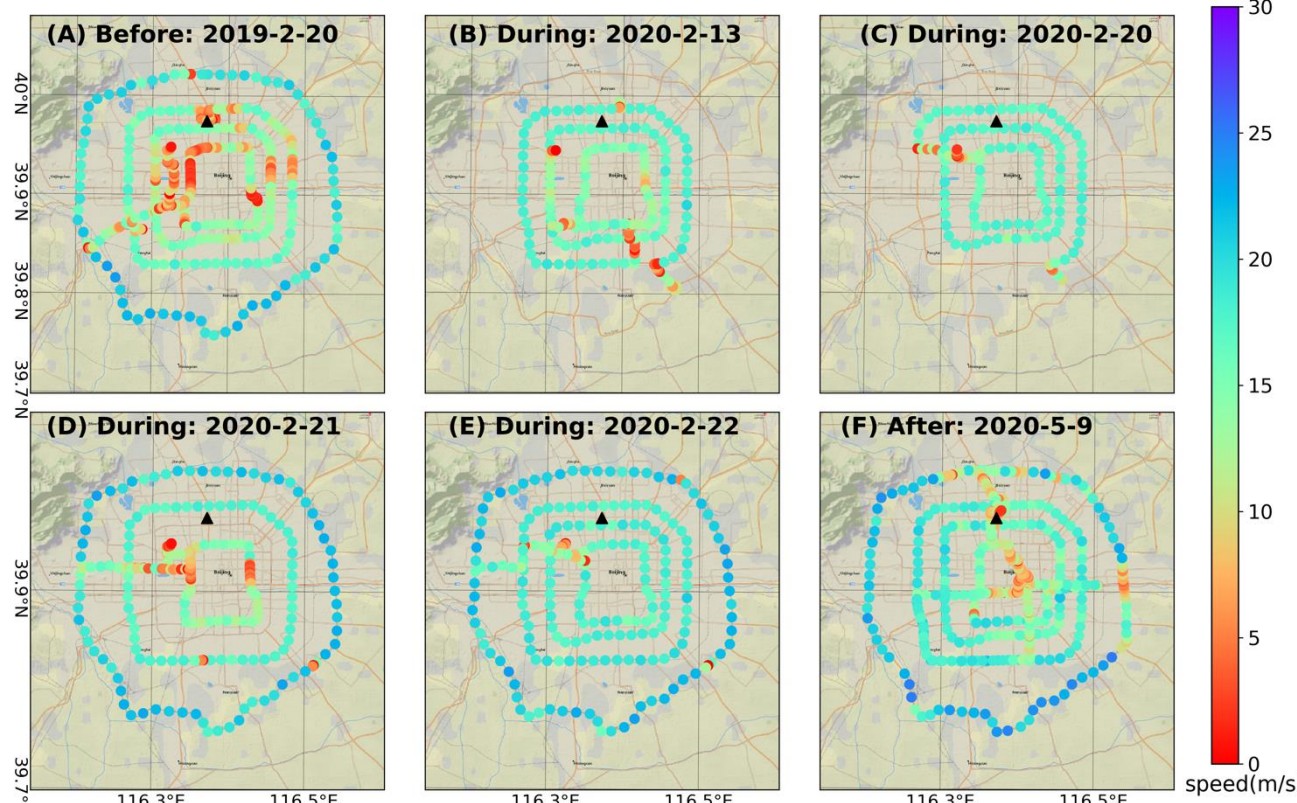

*Figure 10. Speed maps of six trips, ranging from 0 to 30 m/s. One trip (A: 20<sup>th</sup> February 2019) was conducted before the*
*COVID-19 restrictions. Four trips (B-E: 13<sup>th</sup>, 20<sup>th</sup>, 21<sup>st</sup> and 22<sup>nd</sup> February 2020) were conducted during the COVID-19*
*restrictions; one trip (F: 9<sup>th</sup> May 2020) was conducted after the COVID-19 restrictions.*

**Uncertainty analysis:**

In this research, uncertainty mainly existed in the following terms:

(1) Uncertainty from the observation instruments.

In this study, four instruments were adopted for measuring $CO_2$ concentrations: three for on-road observations (a Picarro
G2401, with an accuracy of approximately 0.1 ppm; a LI-COR LI-7810, ~1 ppm; and a low-cost sensor, no more than 5 ppm)
and one for the IAP tower observation (Picarro G2301, ~ 0.1 ppm). During analysis, both the proposed enhancement method
and the $CO_2$ concentration/enhancements of different trips were compared using linear analysis (addition/subtraction).
Therefore, the enhancement uncertainties from the observation instruments were: ~0.2 ppm for BC, ~1.1 ppm for DC, less
than 5.1 ppm for AC, ~1.3 ppm for comparing BC and DC, and less than 6.2 ppm for comparing DC and AC. Note that the
standard deviations shown in Table 4 mainly presented $CO_2$ concentration fluctuations within specific periods and on certain
roads and uncertainty from instruments (relatively small).

(2) The IAP tower $CO_2$ concentration was used as the background from Beijing.

In this study, the IAP tower data were adopted as the urban background $CO_2$ concentration in Beijing. Its measurement
footprint was influenced by two factors: wind speed/direction and air intake height. For wind speed/direction, in Beijing, the
main wind directions were northwest (winter) and southeast (summer) (Cheng et al., 2018). Generally, high-level data have a
large footprint and good representativeness. For example, Cheng *et al.* (2018) showed that $CO_2$ data recorded at 280 m
height have an average fetch of ~17 km, which covers a major part of the city; data collected at 80 m height have an average
fetch of ~8 km; data collected at 8 m height may have an average fetch of only ~230 m; and the fetch at the surface (2 m)
may be smaller. Therefore, there are two uncertainties. The first is the height variation during the observation trips. Due to
the data availability and for comparison consistency, we chose the lower- and surface-level data. According to Cheng *et al.*

(2018), the $CO_2$ concentration at the 80 m height is ~15 ppm higher than that at the 8 m height. Therefore, if this difference between the lower level and surface level was added, the BC enhancement would increase (~15 ppm), which means that the DC enhancement would be even lower (~56 ppm) than the BC enhancement. The other is the difference between the surface level data and 280-metre height data in different seasons. According to Cheng *et al.* (2018), the monthly averaged $CO_2$ showed a relatively stable difference among the different heights: the $CO_2$ at the lower level was approximately 40 ppm higher than that at 280-metrs in February and approximately 30 ppm higher in May. The AC enhancement should increase 10 ppm additionally, which means that the DC enhancement would be even lower (~36 ppm) than the AC enhancement. Considering these uncertainties, the results support our hypothesis.

(3) Influences of vegetation sinks and natural changes.

To understand the $CO_2$ variability impacted by natural sinks (especially for vegetation), we used the dynamic vegetation and terrestrial carbon cycle model VEGAS (Zeng *et al.,* 2014) to simulate the terrestrial biosphere-atmosphere flux (Fta) in Beijing during 2000-2020 (SFigure 3). The model was run at a 2.5×2.5-degree resolution from 1901 to June 2020, forced by observed climate variables, including monthly precipitation and hourly temperature. Although precipitation and temperature in 2020 were higher than the climatology (average of last 20 years), the difference between the Fta in 2020 and the average was within one standard deviation. This suggests that the Fta in 2020 was not obviously unusual compared to that over the last 20 years. We also analysed the $CO_2$ concentration at the Shangdianzi station in the Beijing rural region, which is one of the three WMO/GAW regional stations in China, to determine the $CO_2$ background variation (Fang et al., 2016). The results (SFigure 4) showed that the background $CO_2$ concentration variation mainly induced by natural factors from February to May was only approximately 5 ppm. However, these two factors (vegetation flux and natural changes) both indicate areas far larger than Beijing urban areas. Because the location of the IAP tower and the tracks of the on-road observations are both in urban Beijing and we used the enhancement method, these factors were reduced.

(4) When data were collected, especially when switching between lower and upper levels, a large amount of data was lost. However, because the data gaps were evenly distributed and the IAP tower $CO_2$ concentrations were relatively stable, we assumed that it would not affect the final statistical results.

(5) In this study, our on-road observations did not have a fixed route or beginning/ending time, which means that the observations on different dates represented different roads. Therefore, we analysed a wide time range of observations (rush hours, working hours or whole days), which may have also caused uncertainty.

**Conclusion**

The $CO_2$ emission reduction caused by COVID-19 restrictions is an opportunity to test our ability to collect $CO_2$ observations in urban areas. In this study, we chose on-road $CO_2$ concentrations as the target, because ground transportation is the main source of $CO_2$ in urban areas and was remarkably influenced by policy restrictions due to the COVID-19 pandemic. We conducted six on-road observations in Beijing, including one trip before COVID-19 restrictions, in February 2019; four trips during COVID-19 restrictions, in February 2020; and one trip in May 2020, after COVID-19 restrictions had been eased. The results showed that on-road $CO_2$ concentrations were strongly affected by traffic emissions and weather. However, the enhancement metric, which was the difference in the on-road $CO_2$ concentration and the city "background", reduced the impact of background $CO_2$ fluctuations. The results showed that for DC, the total average $CO_2$ enhancements of the four trips were 41 (±1.3) ppm and 26 (±6.2) ppm lower than those for BC and AC, respectively. Detailed analysis showed that this reduction commonly existed on all road types during the same time period (rush hours/working hours). For the DC trips, there was no significant difference during work hours between weekdays and weekends. The enhancements during rush hours were much higher than those during working hours, and compared with the enhancement reduction during rush hours for BC, that for DC was more obvious. Our findings, which show a clear decrease for DC compared with BC and AC, are consistent with the COVID-19 restrictions, which may be direct evidence of reductions in $CO_2$ concentrations and carbon emissions. On-road $CO_2$ observations are an effective way to understand and analyse the urban carbon $CO_2$ concentration distribution and variation and should be regularly and more frequently conducted in future work. The

development and successful application of the miniaturized and low-cost $CO_2$ monitoring instruments used in this study (Khan et al., 2012;Shusterman et al., 2016;Martin et al., 2017;Mueller et al., 2020;Bao et al., 2020) will greatly aid in the collection of on-road observations and even high-density network observations and play a key role in future urban carbon observations.

**Author contributions**

Pengfei Han, Bo Yao and Ning Zeng conceived and designed the study. Di Liu summarized the results and wrote the draft of the paper. Wanqi Sun and Pengfei Han designed and conducted the on-road observations. Pucai Wang provided the IAP tower observation data. Ke Zheng, Zhiqiang Liu, Han Mei and Qixiang Cai helped to collect, process and analyse data.

**Competing interests.**

The authors declare that they have no conflicts of interest.

**Acknowledgements:**

This work was supported by the National Key R&D Program of China (No. 2017YFB0504000). Special thanks are given to Zhe Hu, Zhimin Zhang and Xiaoli Zhou for collecting the data and conducting the observations.

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
