# Peer review of "Observed decreases in on-road CO2 concentrations in Beijing during"

_Atmospheric Chemistry and Physics, 2020_

## Referee Comment (RC1) · Anonymous Referee #1 · 3 Nov 2020

The MS mainly deals with atmospheric CO2 concentrations measured during 6 on-road observation trips in Beijing, China using mobile platforms before (1 trip), during (3 trips) and after (2 trips) the local COVID-19 restrictions. The topic belongs to the scope of the ACP, it is timely and of research interest. The general levels of the evaluations and discussion, however, should be largely improved. Due to this severe contradiction, which is documented in the comments listed below as examples, the evaluation of the MS cannot be performed unambiguously.

Major comments

1. To reduce the weather and background impacts on the atmospheric concentrations, the authors selected the days which were similar to each other as far as the local weather is concerned. They used reality photos collected from the IAP tower, PM2.5

mass concentrations and WS data for this purpose. Looking at the photos and PM2.5 data, it seems, however, that some days were rather different from the others. The PM2.5 mass concentrations, for instance, changed from 6 to 169 microg m-3. Do these conditions really represent similar weather? (Furthermore, can the latter case indeed be classified as "Light polluted day"?) In addition, one can only wonder why the authors did not use visibility data (possibly available from the AP tower as well) instead of photos, which are demonstrative character only.

2. More importantly, the planetary boundary layer height (PBLH) - which is an important property that can affect the actual concentration of pollutants emitted from surface sources - was not taken into consideration and discussed. The same arguments partially hold for precipitation (and for vegetation activity over the months). All these should be included and addressed in detail in the revised MS.

3. The number of trips (1/3/2) was rather limited. The authors should discuss the representativity of their results and conclusions. Of the 6 trips, there were 5 trips performed on weekdays and 1 trip on weekend. The authors should clarify their statement that "During COVID-19, there was no significant difference between weekdays and weekends." Could this sentence be made more specific or is the number of trips sufficient for the conclusion.

4. L78-80: The authors state "... the enhancement, which calculates the difference in the $CO_2$ concentration between urban and rural background observations, could effectively reduce the influence of background $CO_2$ concentration fluctuations to analyze $CO_2$ concentration characteristics in urban areas...". The IAP tower is, however, located in the city, and there are no arguments why it should be considered as the background environment. The authors refer to its values only as baseline concentrations, which were obtained at the lower or surface levels. One can wonder if the area of the trips and the site of the tower (in particular at lower levels) are influenced by the same environmental conditions. In addition, what is the prevailing wind direction in the area?

Minor comments

5. The MS is extremely difficult to read which detracts from its values. It should be better organized, some strange citation practice (e.g. L58: ... from Le Quere et al.(Le Quere et al., 2020)), the rounding off strategy (e.g. L51: ... emissions dropped abruptly by 53.4%), oversophisticated formulations or non-consistent presentations (e.g. Fig. 1, panels C and D: color coding/line representation reversed, it contains dashed and not dotted lines as specified), spelling mistakes (L61: ... difficult to detect a decrease in the urban CO2 concentration decrease directly) and frequent redundant repetitions should be carefully revisited and corrected. This all implies that the authors should have paid more attention to finalizing their MS.

––––––––––––––––––––––––––

---

## Referee Comment (RC2) · Anonymous Referee #2 · 10 Dec 2020

The authors adopt the idea of investigating urban CO2 enhancements, by reducing the concept of a flux budget approach to a horizontal concentration gradient measurement that typically enters advection flux calculations. The approach would be more meaningful if it was at least combined with a wind field analysis (see: doi: 10.1016/j.atmosenv.2010.02.026, doi: 10.5194/amt-8-3745-2015) and analysis of horizontal advection, rather than using the quite qualitative and subjective argument of 'similar weather patterns'. As it stands, my concern is that the presented findings are rather qualitative. It would be expected that CO2 emissions during the lockdown should drop due to reduced traffic loads. This has been observed and documented before for China. I could see the methodology combined with a more in depth budgetary analysis appropriate for ACP (see references above), but concerns outlined below should be

[Figure]

addressed.

A key uncertainty of the presented analysis is that the obtained enhancement comparisons during the lockdown are quite unspecific for quantifying changes of urban $CO_2$ emissions. They are certainly not representative of a city scale change, because measurements are biased towards road traffic. On the other hand the analysis can not tease out exclusive changes due to road traffic either, because these measurements will almost certainly be influenced by other urban $CO_2$ combustion sources (e.g. in the residential, public and commercial sectors) that might have changed quite differently during the lockdown period (e.g. doi: 10.1038/s41558-020-0797-x; doi: 10.1038/s41467-020-18922-7). As such it leaves the reader with a rather vague number for the obtained $CO_2$ decrease during the lockdown. The finding that $CO_2$ changes during a lockdown is not really something new for China. While qualitatively it makes sense that traffic reduced urban $CO_2$ emissions, a quantitative value can not be easily justified, because the observed changes are also influenced by other processes, that are only poorly constrained by the analysis. In particular what is the influence of urban heating devices and other combustion sources, where some studies have suggested increased demand, others have suggested decreased demand during the lockdown? The lack of a weekend weekday effect seems to corroborate this concern. The comparison before, during and after lockdown is qualitative at best, and it is not clear why this hasn't been combined with a more thorough analysis of advection.

Specific comments: Line 19: what about the biological sink?

Line 24: Grammar: means ? - 'the' onroad. . ..

Line 31-32: The enhancement ratio per se does not eliminate the impact of weather. Furthermore even under similar meteorological conditions, turbulent transport in the street canyon could significantly influence enhancement ratios and ambient concentrations. This statement is therefore not substantiated by the presented approach.

Line 38: The cited reference of le Quere does not investigate the dynamics of the

pandemic. I suggest to use a more appropriate reference (see WHO literature on that).

Line 45: change to 'industrial production'

Line 57: change to 'urban areas' .

Line 57: specify what you mean by weather changes...

Line 64: be more specific about turbulence here – do you mean turbulent mixing and stable conditions? For a statement like that you should specify quantitative parameters such as the Monin Obukhov length or similar parameters to backup your opinion.

Line 66: change to 'global emission reductions'.... and ... 'Despite global emission reductions...'

Line 75: what means transects and communities?

Line 77: what do you mean by diffusion condition?

Line 80 cc: While enhancement ratios could reduce background $CO_2$ variability there are many more factors in urban areas that could influence the quantitative change of $CO_2$. For example vegetation can reduce $CO_2$ concentrations. I acknowledge that this might not be a huge factor in winter, but the generalization made here is rather bold and certainly an oversimplification of the problem. If the lockdown in Beijing occurred in summer there would definitely be a big influence from the vegetative sink of $CO_2$.

Line 84 cc and figure 1: you start discussing results including a figure in the introduction without introducing the methodology before. This paragraph should be rearranged and moved to the results section.

Line 100: May 9th 2020 serves as a post lockdown reference day, but it was a Saturday – so what justification is there to use this day for a post lockdown reference day?

Line 108: what are reality photos?

Line 143: The reference should actually read Sun et al., (2019) and not Sun et al. (Sun

et al., 2019)

Line 205-235: It is not clear what statistics was used to determine whether the results are statistically significant. A proper error propagation and uncertainty analysis should be outlined. It is unclear why no error bars are provided here. When I see a value of 477 ppm I wonder how accurate this number really is, given, that some of the data are based on cheap air quality sensor measurements. This is a major weakness of the present form of the manuscript. A rigorous scientific analysis of errors (systematic and random) needs to be included in order to validate the presented results. It would also be good to see the magnitude of natural variability; how much of the observed variability is due to instrument noise and detection limits, and how much is natural variability? This is a fundamentally missing piece of the analysis. Without such an analysis I have doubts that the presented conclusions are justifiable.

Table 3: There is no statistical analysis presented here.

Table 4: there could be a statistical significance test applied to these data. In this context the section on uncertainty analysis is somewhat apart from the rest, and fails to apply a rigorous mathematical approach to analyzing such kind of data in a statistical sense. I would strongly encourage to improve this part of the manuscript.

Line 379 cc (conclusion): the sentence is incomplete and/or grammatically wrong.

Line 385: there is no significant WE/WD effect during COVID, which is a surprise – how different were traffic flow patterns between WE and WD? I suggest to look at typical WE/WD traffic count data and compare these with CO2 results. If indeed there was no WE/WD effect for traffic count data, one would accept this finding. Otherwise it suggests that one needs to be cautious when extrapolating CO2 enhancement measurements along roads to traffic related changes alone.

---

## Author Comment (AC1) · 3 Feb 2021

Reply to reviewer's comments:

We appreciate the careful evaluation and many important comments from the reviewers. We made additional analyses and major revisions to the paper, including the following: 1) Uncertainty analysis for observation instruments and significance test 2) The impact of the biological (vegetation) sink. 3) We rewrote and reorganized the on-road CO2 and enhancement results and statement.

Below, we describe these changes in detail and address comments and suggestions point-by-point.

Please see the supplement to this comment for figure details and blue font response.

[Figure]

Reply to Reviewer 1:

The MS mainly deals with atmospheric CO2 concentrations measured during 6 on-road observation trips in Beijing, China using mobile platforms before (1 trip), during (3 trips) and after (2 trips) the local COVID-19 restrictions. The topic belongs to the scope of the ACP, it is timely and of research interest. The general levels of the evaluations and discussion, however, should be largely improved. Due to this severe contradiction, which is documented in the comments listed below as examples, the evaluation of the MS cannot be performed unambiguously. Major comments 1. To reduce the weather and background impacts on the atmospheric concentrations, the authors selected the days which were similar to each other as far as the local weather is concerned. They used reality photos collected from the IAP tower, PM2.5 mass concentrations and WS data for this purpose. Looking at the photos and PM2.5 data, it seems, however, that some days were rather different from the others. The PM2.5 mass concentrations, for instance, changed from 6 to 169 microg m-3. Do these conditions really represent similar weather? (Furthermore, can the latter case indeed be classified as "Light polluted day"?) In addition, one can only wonder why the authors did not use visibility data (possibly available from the AP tower as well) instead of photos, which are demonstrative character only. Thank you for your comments and suggestions. We may have misled you and the readers. We reorganized the Results section in the revised manuscript. We conducted six observation trips, one trip for before COVID-19 restrictions (BC), four trips for during COVID-19 restrictions (DC) and one trip for after COVID-19 restrictions (AC). We only used 2 DC trips (on 21st and 22nd February 2020) from 4 DC trips, which we ensured that the weather conditions were as similar as possible. Therefore, we compared the CO2 concentrations on the roads using these 4 trips (1 trip for BC, 2 trips for DC and 1 trip for AC) with similar weather conditions. The other two trips were also plotted and are shown in the Results section, but they were not used for comparison. Although these 4 trips (2 DC trips with the most similar weather conditions, 1 BC trip and 1 AC trip) were labelled as having the most similar weather conditions, their weather conditions were still different (from table 1, the PBLHs and wind speeds were

not exactly the same), and they were still impacted by background $CO_2$ fluctuations. Therefore, to further reduce these impacts, we used the enhancement method. We found that this method works for both the 2 clear-day DC trips (labelled as having the most similar weather conditions) and the 2 polluted-day DC trips (on 13th and 20th February 2020). February 13th (PM2.5 equalled 169), was heavy pollution day. We have revised accordingly. Thanks! We did not find visibility data for the IAP tower.

2. More importantly, the planetary boundary layer height (PBLH) - which is an important property that can affect the actual concentration of pollutants emitted from surface sources - was not taken into consideration and discussed. The same arguments partially hold for precipitation (and for vegetation activity over the months). All these should be included and addressed in detail in the revised MS. Thank you for your suggestions. We added PBLH data from GFS in Table 1 and added a discussion in the "Methods and Data" and "Results" sections. To understand the $CO_2$ variability impacted by natural sinks (especially for vegetation), we used the dynamic vegetation and terrestrial carbon cycle model VEGAS (Zeng et al., 2014) to simulate the terrestrial biosphere-atmosphere flux (Fta) in Beijing during 2000-2020 (SFigure 3). The model was run at a 2.5×2.5-degree resolution from 1901 to June 2020, forced by observed climate variables, including monthly precipitation and hourly temperature. Although precipitation and temperature in 2020 were higher than the climatology (average of last 20 years), the difference between the Fta in 2020 and the average was within one standard deviation. We also analysed the $CO_2$ concentration at the Shangdianzi station in the Beijing rural region, which is one of the three WMO/GAW regional stations in China, to determine the $CO_2$ background variation (Fang et al., 2016). The results (SFigure4) showed that the background $CO_2$ concentration variation mainly induced by natural factors from February to May was only approximately 5 ppm.

However, these two factors (vegetation flux and natural changes) both indicate areas far larger than Beijing urban areas. Because the location of the IAP tower and the tracks of the on-road observations are both in urban and we used enhancement method,

these factors were reduced.

3. The number of trips (1/3/2) was rather limited. The authors should discuss the representativity of their results and conclusions. Of the 6 trips, there were 5 trips performed on weekdays and 1 trip on weekend. The authors should clarify their statement that "During COVID-19, there was no significant difference between weekdays and weekends." Could this sentence be made more specific or is the number of trips sufficient for the conclusion. Thank you for your suggestions. We agreed with your suggestion, and we revised the statement as "during COVID-19 restrictions, there was no difference between weekdays and weekends during working hours (9:00-17:00 local standard time, LST)." We also added a traffic flow map for the 4 DC trips (SFigure 2) during working hours to validate our statement. The map shows that during the working hours of the 4 DC trips, the traffic conditions in Beijing were all smooth. As representatives of the results and conclusions, we conducted six trips in this study, which covered both weekdays and weekends, both clear days and polluted days and both rush hours and working hours. We obtained robust signals when we used the enhancement method. When we analysed the results, we categorized the results according to weekdays/weekends, working/rush hours and ring roads/other roads to ensure that the results were comparable. 4. L78-80: The authors state ". . . the enhancement, which calculates the difference in the $CO_2$ concentration between urban and rural background observations, could effectively reduce the influence of background $CO_2$ concentration fluctuations to analyze $CO_2$ concentration characteristics in urban areas. . .". The IAP tower is, however, located in the city, and there are no arguments why it should be considered as the background environment. The authors refer to its values only as baseline concentrations, which were obtained at the lower or surface levels. One can wonder if the area of the trips and the site of the tower (in particular at lower levels) are influenced by the same environmental conditions. In addition, what is the prevailing wind direction in the area? Thank you for your suggestions. First, the 280-metre level data from the IAP tower have good representativity because they have an average fetch of approximately 17 km, which covers a major part of Beijing's urban areas. Unfortunately, 280-metre

level data were missed due to the instrument (sampling pump) malfunctions, and we obtained only surface and lower-level data. Second, the monthly averaged $CO_2$ from different levels according to Cheng et al. (2018) showed a relatively stable difference among different heights (shown below). The results showed that the 8-metre level was approximately 40 ppm higher than 280 metres level in February and approximately 30 ppm higher in May. Therefore, the AC enhancement should increase by 10 ppm additionally. And this result supports with our hypothesis.

Third, there are no large emissions sources near the IAP tower. The prevailing wind direction during these trips was northwest.

5. The MS is extremely difficult to read which detracts from its values. It should be better organized, some strange citation practice (e.g. L58: . . . from Le Quere et al.(Le Quere et al., 2020)), the rounding off strategy (e.g. L51: . . . emissions dropped abruptly by 53.4%), oversophisticated formulations or non-consistent presentations (e.g. Fig. 1, panels C and D: color coding/line representation reversed, it contains dashed and not dotted lines as specified), spelling mistakes (L61: . . . difficult to detect a decrease in the urban $CO_2$ concentration decrease directly) and frequent redundant repetitions should be carefully revisited and corrected. This all implies that the authors should have paid more attention to finalizing their MS. Thank you! We have carefully edited the manuscript and corrected all the instances you pointed out and several other places. We also reorganized and rewrote the Results section. Regarding the strange citation practice, a technical error occurred when we used Endnote software to automatically generate citations in MS Word document, we have carefully checked and revised the manuscript.  References: Cheng, X. L., Liu, X. M., Liu, Y. J., and Hu, F.: Characteristics of $CO_2$ Concentration and Flux in the Beijing Urban Area, Journal of Geophysical Research-Atmospheres, 123, 1785-1801, 10.1002/2017jd027409, 2018. Fang, S. X., Tans, P. P., Dong, F., Zhou, H., and Luan, T.: Characteristics of atmospheric $CO_2$ and $CH_4$ at the Shangdianzi regional background station in China. Atmospheric Environment, 131, 1-8, 2016. Kutsch, W.,

[Printer-friendly version]{.underline}

[Discussion paper]{.underline}

Vermeulen, A., Karstens, U., 2020. Finding a hair in the swimming pool: the signal of changed fossil emissions in the atmosphere. https://www.icos-cp.eu/ event/917. Liu, Z., Ciais, P., Deng, Z., Lei, R., Davis, S. J., Feng, S., Zheng, B., Cui, D., Dou, X., Zhu, B., Guo, R., Ke, P., Sun, T., Lu, C., He, P., Wang, Y., Yue, X., Wang, Y., Lei, Y., Zhou, H., Cai, Z., Wu, Y., Guo, R., Han, T., Xue, J., Boucher, O., Boucher, E., Chevallier, F., Tanaka, K., Wei, Y., Zhong, H., Kang, C., Zhang, N., Chen, B., Xi, F., Liu, M., Bréon, F. M., Lu, Y., Zhang, Q., Guan, D., Gong, P., Kammen, D. M., He, K., Schellnhuber, H. J.: Near-real-time monitoring of global CO2 emissions reveals the effects of the COVID-19 pandemic. Nat Commun. 2020 Oct 14;11(1):5172. Ott, L., Peters, G., Meyer, A., 2020. Special virtual panel: Covid-19 and its impact on global carbon emissions. https://carbon.nasa.gov/policy_speaker_28052020.html. Zeng, N., Zhao, F., Collatz, G. J., Kalnay, E., Salawitch, R. J., West, T. O., and Guanter, L.: Agricultural Green Revolution as a driver of increasing atmospheric CO2 seasonal amplitude, Nature, 515, 394-+, 10.1038/nature13893, 2014.

Please also note the supplement to this comment:
https://acp.copernicus.org/preprints/acp-2020-966/acp-2020-966-AC1-supplement.pdf
* * *

---

## Author Comment (AC2) · 3 Feb 2021

Reply to reviewer's comments:

We appreciate the careful evaluation and many important comments from the reviewers. We made additional analyses and major revisions to the paper, including the following: 1) Uncertainty analysis for observation instruments and significance test 2) The impact of the biological (vegetation) sink. 3) We rewrote and reorganized the on-road CO2 and enhancement results and statement.

Below, we describe these changes in detail and address comments and suggestions point-by-point.

Please see the supplement to this comment for figure details and blue font response.

[Figure]

Reply to Reviewer 2:

The authors adopt the idea of investigating urban CO2 enhancements, by reducing the concept of a flux budget approach to a horizontal concentration gradient measurement that typically enters advection flux calculations. The approach would be more meaningful if it was at least combined with a wind field analysis (see: doi: 10.1016/j.atmosenv.2010.02.026, doi: 10.5194/amt-8-3745-2015) and analysis of horizontal advection, rather than using the quite qualitative and subjective argument of 'similar weather patterns'. As it stands, my concern is that the presented findings are rather qualitative. It would be expected that CO2 emissions during the lockdown should drop due to reduced traffic loads. This has been observed and documented before for China. I could see the methodology combined with a more in-depth budgetary analysis appropriate for ACP (see references above), but concerns outlined below should be addressed. Thank you for your suggestion. The purpose of this paper is to provide evidence of on-road CO2 concentration reductions in Beijing influenced by traffic emissions that were substantially decreased due to the lockdown and stay-at-home order. Using budgetary analysis (such as the mass balance method) to calculate CO2 emissions is beyond the scope of this manuscript. In addition, the budgetary analysis (such as the mass balance method) is good at calculating the emissions for an entire city, not only ground transportation emissions. The trajectories of these mobile observations (such as aircraft) should be around the whole city and far away from emission sources, to obtain the downwind CO2 concentration and to further calculate the emissions for the whole city. However, the mobile observations in this study were conducted on roads, and the results largely reflected the on-road emissions information. Therefore, it is not suitable to use budgetary analysis (such as the mass balance method) in this study. Moreover, we did not have enough weather data to implement budgetary analysis (such as the mass balance method) right now. Thanks again.

A key uncertainty of the presented analysis is that the obtained enhancement comparisons during the lockdown are quite unspecific for quantifying changes of urban

CO2 emissions. They are certainly not representative of a city scale change, because measurements are biased towards road traffic. On the other hand the analysis can not tease out exclusive changes due to road traffic either, because these measurements will almost certainly be influenced by other urban CO2 combustion sources (e.g. in the residential, public and commercial sectors) that might have changed quite differently during the lockdown period (e.g. doi: 10.1038/s41558-020-0797-x; doi: 10.1038/s41467-020-18922-7). As such it leaves the reader with a rather vague number for the obtained CO2 decrease during the lockdown. The finding that CO2 changes during a lockdown is not really something new for China. While qualitatively it makes sense that traffic reduced urban CO2 emissions, a quantitative value cannot be easily justified, because the observed changes are also influenced by other processes, that are only poorly constrained by the analysis. In particular what is the influence of urban heating devices and other combustion sources, where some studies have suggested increased demand, others have suggested decreased demand during the lockdown? The lack of a weekend weekday effect seems to corroborate this concern. The comparison before, during and after lockdown is qualitative at best, and it is not clear why this hasn't been combined with a more thorough analysis of advection. Thank you for your suggestion. The aim of this study is to capture the on-road observation CO2 decrease signal induced by ground transportation reduction due to COVID-19 restrictions. We agree with you that this article is a more qualitative rather than quantitative study. However, the article is still of great value. Because the main contribution and significance of this paper is that we observed a clear CO2 concentration decrease induced by ground transportation emission reduction due to COVID-19 restrictions. As we discussed in the Introduction section, although the global emission reduction due to COVID-19 restrictions is very huge, it is very difficult to observe the CO2 concentration decrease from ground-based CO2 concentration monitoring (Kutsch et al., 2020; Ott et al., 2020). Therefore, choosing a suitable research target is the key to observing the CO2 decrease signal.

We agree with you that the on-road observations were inevitably affected by other emissions (e.g. commercial and residential) along the road. However, it is true that ground transportation emissions are the most important signal in on-road observations. Additionally, the enhancement method adopted in this study would reduce the background $CO_2$ concentration fluctuations. "The lack of a weekend weekday effect seems to corroborate this concern". We believe that the lack of a weekend/weekday effect is a reflection of the COVID-19 restrictions and work times (9:00 – 17:00 LST). This is because of the limitation of ground transportation emissions rather than other emissions (e.g., commercial and residential emissions). During DC evening rush hours, the enchantments were as high as those during BC evening rush hours (Figure 5A, 5B and 5D, 2nd Ring Road). We do not have enough weather data to support us do advection analysis. According to Cheng et al., (2018), the monthly average $CO_2$ concentrations from the 8 m and 280 m height levels differ by a relatively consistent amount. Because there are no strong emission sources, such as factories, near the IAP tower, the $CO_2$ concentration difference is mainly caused by diffusion. Specific comments: Line 19: what about the biological sink? Thank you for your careful review. We added "biological sink" in Line 19 (clean version document). Line 24: Grammar: means ? - 'the' onroad. . .. Thank you for your careful review. Revised accordingly. Line 31-32: The enhancement ratio per se does not eliminate the impact of weather. Furthermore even under similar meteorological conditions, turbulent transport in the street canyon could significantly influence enhancement ratios and ambient concentrations. This statement is therefore not substantiated by the presented approach. Thank you for your suggestions. We have changed "eliminate" to "reduce". It indeed does not completely eliminate impacts from weather; however, it could reduce the influence. Line 38: The cited reference of le Quere does not investigate the dynamics of the pandemic. I suggest to use a more appropriate reference (see WHO literature on that). Thank you for your careful review. We used the WHO COVID-19 situation reports. Line 45: change to 'industrial production' Thank you for your careful review. Revised accordingly.

Line 57: change to 'urban areas' . Thank you for your careful review. Revised accordingly.

Line 57: specify what you mean by weather changes. . . Thank you for your suggestions, and we added the explanation as 'for example, high wind speed accelerates the mixing and diffusion of $CO_2$' in Line60-61 (clean version document). Line 64: be more specific about turbulence here – do you mean turbulent mixing and stable conditions? For a statement like that you should specify quantitative parameters such as the Monin Obukhov length or similar parameters to backup your opinion. Thank you for your suggestions. We mean "stable conditions", and we added "in which the planetary boundary layer heights (PBLH) were only $\sim$600 m" in Line68 (clean version document) to back up our opinion. Line 66: change to 'global emission reductions'. . .. and . . . 'Despite global emission reductions. . .' Thank you for your careful review. Revised accordingly. Line 75: what means transects and communities? Thank you for your careful review. We added the detailed explanations for the transects in line80 (clean version document) as 'for instance, on-road $CO_2$ concentration distributions were presented as transects in urban areas along routes'. In addition, we removed the ambiguous word 'communities'. Line 77: what do you mean by diffusion condition? Thank you for your careful review. We added the detailed explanation in line81 (clean version document), as 'for example, wind speed which is directly associated with $CO_2$ mixing and dilution'. Additionally, the ambiguous word 'diffusion condition' was removed. Line 80 cc: While enhancement ratios could reduce background $CO_2$ variability there are many more factors in urban areas that could influence the quantitative change of $CO_2$. For example vegetation can reduce $CO_2$ concentrations. I acknowledge that this might not be a huge factor in winter, but the generalization made here is rather bold and certainly an oversimplification of the problem. If the lockdown in Beijing occurred in summer there would definitely be a big influence from the vegetative sink of $CO_2$. Thank you for your suggestions. We added a discussion about the "influence of vegetation sinks". To understand the $CO_2$ variability impacted by natural sinks (especially for vegetation), we used the dynamic vegetation and terrestrial carbon cycle model VEGAS (Zeng et al., 2014) to simulate the terrestrial biosphere-atmosphere flux (Fta) in Beijing during 2000-2020 (SFigure 3). The model was run at a $2.5\times2.5$-degree resolution from 1901

to June 2020, forced by observed climate variables, including monthly precipitation and hourly temperature. Although precipitation and temperature in 2020 were higher than the climatology (average of last 20 years), the difference between the Fta in 2020 and the average was within one standard deviation. This suggests that the Fta in 2020 was not obviously unusual compared to that over the last 20 years. We also analysed the $CO_2$ concentration at the Shangdianzi station in the Beijing rural region, which is one of the three WMO/GAW regional stations in China, to determine the $CO_2$ background variation (Fang et al., 2016). The results (SFigure 4) showed that the background $CO_2$ concentration variation mainly induced by natural factors from February to May was only approximately 5 ppm.

In addition, the IAP tower and on-road observations were conducted in urban areas, and vegetation in urban areas is much less than that in rural areas. Therefore, the impacts of vegetation sinks could be reduced by using enhancement. Line 84 cc and figure 1: you start discussing results including a figure in the introduction without introducing the methodology before. This paragraph should be rearranged and moved to the results section. Thank you for your careful review. Figure1 shows the background for the on-road observations, and the data were collected from online news and publications (Liu et al., 2020). We believe that it is not appropriate to move this to the Result section. In addition, we rearranged this figure in the Supplementary Material. Thank you. Line 100: May 9th 2020 serves as a post lockdown reference day, but it was a Saturday – so what justification is there to use this day for a post lockdown reference day? Thank you for your careful review. At that time, we considered the feasibility of the experiment, including the observation instruments, urban traffic restrictions and personnel arrangements. We have only this one observation after the lockdown. This is indeed a shortcoming of our experiment, and we will improve on this in the future, which is also written in our conclusion. Line 108: what are reality photos? Thank you for your careful review. We changed reality photos to "Real-time panoramic photographs".

Line 143: The reference should actually read Sun et al., (2019) and not Sun et al. (Sun

et al., 2019) Thank you for pointing this out! This was a technical error on our side: it happened when we used Endnote to automatically generate citations in MS word document, we have carefully checked and revised the manuscript. Line 205-235: It is not clear what statistics was used to determine whether the results are statistically significant. A proper error propagation and uncertainty analysis should be outlined. It is unclear why no error bars are provided here. When I see a value of 477 ppm I wonder how accurate this number really is, given, that some of the data are based on cheap air quality sensor measurements. This is a major weakness of the present form of the manuscript. A rigorous scientific analysis of errors (systematic and random) needs to be included in order to validate the presented results. It would also be good to see the magnitude of natural variability; how much of the observed variability is due to instrument noise and detection limits, and how much is natural variability? This is a fundamentally missing piece of the analysis. Without such an analysis I have doubts that the presented conclusions are justifiable. Thank you for your careful review. The uncertainty from the measurement instruments was ∼0.1 ppm (from Picarro G2301/G2401), ∼1 ppm (from LI-COR LI-7810) and less than 5 ppm (from the low-cost sensor, K30; also there was a validation of 3.6 ppm in method sector). We added the uncertainty for each observed value in the MS. We also added this information to the Uncertainty analysis section. We also conducted a significance test (Table 4): "The $CO_2$ enhancement for BC was also significantly different from that for DC ($p < 0.05$); however, the difference between the AC and BC enhancements was not significant. This suggests that the decreased $CO_2$ enhancement observed during the COVID-19 restrictions was significantly different from those before and after the COVID-19 restrictions." Four instruments were used in this study and are described in the methods and data sections: 1. Picarro (G2301) in the IAP tower: the precision was ±0.1 ppm, and it was calibrated with standard gas (traced to the World Meteorological Organization, WMO) 4 times each day to ensure that its accuracy was ±0.1 ppm. 2. Picarro (G2401) for on-road observation: the precision was ±0.1 ppm, and it was calibrated with standard gas before departure to ensure that its accuracy was ±0.1 ppm. 3. LI-

COR LI-7810 for on-road observation: the precision was ±1 ppm, and it was calibrated with standard gas before departure to ensure its accuracy was ±1 ppm. 4. Low-cost sensor for on-road observation: the precision was no more than 5 ppm after calibration and environmental correction, and it was calibrated with standard gas before departure to ensure that its accuracy was no more than 5 ppm. We also verified the low-cost sensor, and its data were consistent with Picarro's (RMSE of 3.6 ppm). We added the natural vegetation flux (flux from the terrestrial to atmospheric compartments) variation to the Supplementary Materials to show that the vegetation variation in 2020 was not significantly different from that in the last 20 years. We also adopted the $CO_2$ concentration at the Shangdianzi station, which is one of the three WMO/GAW regional stations in China, to indicate the $CO_2$ background variation (Fang et al., 2016). The $CO_2$ concentration variation induced by natural changes from February to May were approximately 5 ppm. However, these two factors (vegetation flux and natural changes) both indicated an area far larger than that of urban Beijing. Because the location of the IAP tower and the path of the on-road observations are both in urban Beijing, when we used the enhancement method, these factors were reduced.

Table 3: There is no statistical analysis presented here. Thank you for your careful review. We performed a detailed statistical analysis (Table 4), which includes the mean and one standard deviation for different periods and different roads (including a whole trip). Table 3, which was extracted from Table4, presents the main conclusions of this study and is simplified here. We added the instrumental uncertainty in Table 3. Table 4: there could be a statistical significance test applied to these data. In this context the section on uncertainty analysis is somewhat apart from the rest and fails to apply a rigorous mathematical approach to analyzing such kind of data in a statistical sense. I would strongly encourage to improve this part of the manuscript. Thank you for your careful review. We added a statistical significance test, and found that "After a statistical significance test, we found that the $CO_2$ enhancement difference between working times and rush hours for all trips was significant ($p < 0.02$, assuming that $\alpha=0.05$). The $CO_2$ enhancement for BC was also significantly different from that for DC ($p< 0.05$);

[Figure]

however, the difference between the AC and BC enhancements was not significant. This suggests that the decreased $CO_2$ enhancement observed during the COVID-19 restrictions was significantly different from those before and after the COVID-19 restrictions." We believe that the deviation listed in Table 4 is very large, which may mislead you and readers. Considering that the instrumental uncertainties in BC/DC/AC are $\sim$0.2/$\sim$1.1/ less than 6.1 ppm (added in the MS/Uncertainty analysis sector), the deviation here mainly included the on-road $CO_2$ concentration variation. For example, when there is a traffic jam, a large $CO_2$ concentration variation would result in a large deviation. Line 379 cc (conclusion): the sentence is incomplete and/or grammatically wrong. Thank you for your careful review. Revised accordingly. Line 385: there is no significant WE/WD effect during COVID, which is a surprise – how different were traffic flow patterns between WE and WD? I suggest to look at typical WE/WD traffic count data and compare these with $CO_2$ results. If indeed there was no WE/WD effect for traffic count data, one would accept this finding. Otherwise, it suggests that one needs to be cautious when extrapolating $CO_2$ enhancement measurements along roads to traffic related changes alone. Thank you for your careful review. Revised accordingly. We added the traffic flow map during the COVID-19 restrictions for both WE and WD (SFigure 2). During working hours, all ring roads both WE and WD were smooth.

References: Cheng, X. L., Liu, X. M., Liu, Y. J., and Hu, F.: Characteristics of CO2 Concentration and Flux in the Beijing Urban Area, Journal of Geophysical Research-Atmospheres, 123, 1785-1801, 10.1002/2017jd027409, 2018. Fang, S. X., Tans, P. P., Dong, F., Zhou, H., and Luan, T.: Characteristics of atmospheric CO2 and CH4 at the Shangdianzi regional background station in China. Atmospheric Environment, 131, 1-8, 2016. Kutsch, W., Vermeulen, A., Karstens, U., 2020. Finding a hair in the swimming pool: the signal of changed fossil emissions in the atmosphere. https://www.icos-cp.eu/event/917. Liu, Z., Ciais, P., Deng, Z., Lei, R., Davis, S. J., Feng, S., Zheng, B., Cui, D., Dou, X., Zhu, B., Guo, R., Ke, P., Sun, T., Lu, C., He, P., Wang, Y., Yue, X., Wang, Y., Lei, Y., Zhou, H., Cai, Z., Wu, Y., Guo, R., Han, T., Xue, J., Boucher, O., Boucher, E., Chevallier, F., Tanaka, K., Wei, Y., Zhong, H., Kang, C., Zhang, N., Chen, B., Xi,

F., Liu, M., Bréon, F. M., Lu, Y., Zhang, Q., Guan, D., Gong, P., Kammen, D. M., He, K., Schellnhuber, H. J.: Near-real-time monitoring of global CO2 emissions reveals the effects of the COVID-19 pandemic. Nat Commun. 2020 Oct 14;11(1):5172. Ott, L., Peters, G., Meyer, A., 2020. Special virtual panel: Covid-19 and its impact on global carbon emissions. https://carbon.nasa.gov/policy_speaker_28052020.html. Zeng, N., Zhao, F., Collatz, G. J., Kalnay, E., Salawitch, R. J., West, T. O., and Guanter, L.: Agricultural Green Revolution as a driver of increasing atmospheric CO2 seasonal amplitude, Nature, 515, 394-+, 10.1038/nature13893, 2014.

Please also note the supplement to this comment:
https://acp.copernicus.org/preprints/acp-2020-966/acp-2020-966-AC2-supplement.pdf

---

## Author Response (AR1)

[revised manuscript text omitted]

on-road $CO_2$ concentrations, enhancements and trajectories f̶o̶r̶ a̶l̶l̶ t̶r̶i̶p̶s̶. In Figure 6̶7̶A, the I̶A̶P̶ t̶o̶w̶e̶r̶ $CO_2$ concentration
measured at the IAP tower c̶o̶n̶c̶e̶n̶t̶r̶a̶t̶i̶o̶n̶s̶ were r̶e̶l̶a̶t̶i̶v̶e̶l̶y̶ s̶t̶a̶b̶l̶e̶,̶ a̶n̶d̶stable and showed t̶h̶e̶ an d̶i̶f̶f̶e̶r̶e̶n̶c̶e̶approximate 50
ppm difference between trips. The $CO_2$ concentrations at the IAP tower C̶O̶₂̶ c̶o̶n̶c̶e̶n̶t̶r̶a̶t̶i̶o̶n̶s̶ during the first two DC trips t̶h̶e̶
t̶w̶o̶ t̶r̶i̶p̶s̶ d̶u̶r̶i̶n̶g̶ C̶O̶V̶I̶D̶-̶1̶9̶ (13th and 20th February 2020) were ~30 ppm higher than those during the BC and AC trips.
However, the $CO_2$ concentrations c̶o̶n̶c̶e̶n̶t̶r̶a̶t̶i̶o̶n̶s̶ during the other two DC trips (21st and 22nd February 2020) were ~20 ppm
lower than those during the t̶h̶o̶s̶e̶ d̶u̶r̶i̶n̶g̶ t̶h̶e̶BC and AC t̶r̶i̶p̶s̶trips. These "baseline" $CO_2$ concentration fluctuations make
the on-road observations not directly comparable d̶i̶r̶e̶c̶t̶l̶y̶. In Figure 7̶8̶B, the $CO_2$ concentrations show a "double-peak"
pattern, with peaks during w̶i̶t̶h̶i̶n̶ the morning (7:00-9:00) and evening r̶u̶s̶h̶ h̶o̶u̶r̶s̶ (17:00-20:00) rush hours. During the rush
hours, the $CO_2$ concentrations ranged from 500 to 600 ppm, which were approximately 100 ppm higher than the
concentrations during working hours (9:00-17:00). The comparison of BC and AC indicates that the $CO_2$ concentrations
measured on 13th and 20th February 2020 did not significantly decrease during 12:00-17:00. However, the $CO_2$

335 concentrations measured on 21st and 22nd February 2020 were much lower (~50 ppm) than those measured during the BC and AC trips. This difference is consistent with the spatial distribution mentioned before and is most likely due to  background $CO_2$ fluctuations.

340 In Panel C, all DC  enhancements were generally lower than  those  BC and AC enhancements, and the statistics for different time periods are listed in Table 3. However, we  also found small  enhancements  for BC and AC, similar to those for DC. For example,  AC enhancement at  12:00  16:00 was almost the same as that of DC enhancement at that time. By examining the  trip routes (Panel D), we found that during that period, the on-road observation vehicle was not driving on the main ring roads. As another example, BC

345 enhancement at  18:00 indicates that the enhancement decreased in a stepwise manner, also because the vehicle drove on other roads (Panel D).

Table 3.  $CO_2$ enhancement (mean and instrumental uncertainties) for six trips over different periods (ppm)

| Label | Observation date | Weather condition | Total average (07:00-20:00) | Morning rush hours (07:00-09:00) | Midday RUSH hours | Evening rush hours (17:00-20:00) |
|---|---|---|---|---|---|---|
| BC | 2020 (Wed) | Clear | 65 (±0.2) | - | 54 (±0.2) | 100 (±0.2) |
| DC | 2020-2-13 (Thu) | Stable/light pollution | 33 (±1.1) | - | 26 (±1.1) | 55 (±1.1) |
|  |  |  |  |  |  |  |
| |  |  |  |  |  |  |
| |  |  |  |  |  |  |
|  |  |  |  |  |  |  |
| |  | |  |  |  |  |
| |  | |  |  |  |  |

| | | | | | | |
|---|---|---|---|---|---|---|
| | | vy pollution | | | | |
| | 2020-2-20 (Thu) | Stable//light pollution | 16 (±1.1) | - | 16 (±1.1) | - |
| | 2020-2-21 (Fri) | Windy day | 30 (±1.1) | - | 16 (±1.1) | 50 (±1.1) |
| | 2020-2-22 (Sat) | Windy day | 17 (±1.1) | - | 17 (±1.1) | - |
| AC | 2020-5-9 (Sat) | Windy day | 50 (±5.1) | 80 (±5.1) | 46 (±5.1) | - |
| | Total BC-DC | | 41 (±1.3) | - | 35 (±1.3) | 48 (±1.3) |
| | Total AC-DC | | 26 (±6.2) | - | 27 (±6.2) | - |

350

The average of $CO_2$ mean enhancement for the whole BC trip was 65 (±0.2) ppm, and the average for the the evening rush hours (100 ±0.2 ppm) was two times that of for the working hours (54 ±0.2 ppm). This result implies that the increase in vehicle volume in during the evening rush hours leads to large traffic emissions and an increase in the on-road $CO_2$ concentration. For DC, all trips covered the working hours, with a low enhancement of approximately 20 ppm. There was not obvious difference between weekdays and weekends during this periodworking hours, which indicated that there was no "week effect". The reason may be that because the government encouraged people to work remotely at home. Therefore, even on weekdays, according to traffic conditions, the commute volume was low was small(SFigure 2). Among these four trips, two (on 13th and 20th February 2020) covered the evening rush hours with high averaged enhancements of 55 (±1.1) and 50 (±1.1) ppm. Therefore, the total average enhancements averages forof these two trips were higher than those for of the other two trips, which covered only only working hours. For AC, on 9th May 2020, although it was a Saturday, many residents chose to go out of town for the weekends. The morning rush hours still existed, with a high enhancement of 80 (±5.1) ppm, and then during the working hours, the enhancement decreased to 46 (±5.1) ppm.

The comparison of trips showed that the averaged $CO_2$ enhancement for the from 4 whole DC trips was 41 (±1.3) and 26 (±6.2) ppm lower than that from for the BC and AC trips, respectively. Compared to tThe BC trip, the averaged AC enhancement was 15 15 (±5.3) ppm lower than the average BC enhancement. This difference may be caused by two factors: 1) The first relates to "weekly effects".; as previously mentioned; a previous study also suggested that, compared to that during weekdays, the average daily traffic $CO_2$ emissions during weekends in the north part of the fifth Ring Road (LinCui Road - Anli Road, 3 km) decreased by 5% throughout wholein 2014 according to the Motor Vehicle Emission Simulator model (from 131.74 t/d to 126.33 t/d)(Zheng et al., 2020).; 2) until Until 9th May 2020, although there were approximately 30 days without increased increases in COVID-19 cases in Beijing, the city was still under Level-2 response control; social life was recovering, but had not yet completely recovered.

**Analysis of $CO_2$ enhancement on for independent time periods and roads:**
According to the previous analysis, we found that enhancement exhibited a strong correlation with the time (rush or working hours) and road types. Therefore, we statistically analyszed the $CO_2$ enhancements according to the road types and time periods, as shown in Figure 78. In Figure 78A, on 13th and 20th February 2020, the $CO_2$ concentrations on the other, 2nd, and 4th rRing rRoads and all roads were at the same levels as those during the BC and AC trips. However, in Figure 78B, the the enhancement showed that the four DC enhancementsfour trips during COVID-19 were generally lower than those during AC and BC for all road types. Although on the 2nd Ring Road, the DC trips enhancements on 13th and 21st February 2020 were

[revised manuscript text omitted]

In this study, the IAP tower data were adopted as the urban background $CO_2$ concentration in Beijing. Its measurement footprint was influenced by two factors: wind speed/direction and air intake height. For wind speed/direction, in Beijing, the main wind directions were northwest (winter) and southeast (summer) (Cheng et al., 2018). ~However, IAP tower data were collected from different levels, as described in the method section.~ Generally, high-level data ~would~ have a large footprint and good representativeness ~and cover large regions~. For example, Cheng *et al.* (~Cheng et al.,~ 2018) showed that $CO_2$ data recorded at 280 m height ~level~ $CO_2$ ~data~ have an average~d~ fetch of ~17 km, which ~may~ covers a major part of the city; data collected at 80 m height ~level data~ have an average~d~ fetch of ~8 km; data collected at ~and~ 8 m height ~level data~ may have

[revised manuscript text omitted]